# PD²GS: Part-Level Decoupling and Continuous Deformation of Articulated Objects via Gaussian Splatting

**Haowen Wang**[1]   **Xiaoping Yuan**[2]   **Zhao Jin**[3]   **Zhen Zhao**[3]   **Zhengping Che**[3]
**Yousong Xue**[4]   **Jin Tian**[4]   **Yakun Huang**[2]*   **Jian Tang**[3]*
[1]Anhui University    [2]Beijing University of Posts and Telecommunications
[3]Beijing Innovation Center of Humanoid Robotics    [4]Beijing Institute of Architectural Design
wanghaowen@ahu.edu.cn
{mustafa.jin, alex.zhao, z.che, jian.tang}@x-humanoid.com
{xiaopingyuan, ykhuang}@bupt.edu.cn

## Abstract

Articulated objects are ubiquitous and important in robotics, AR/VR, and digital twins. Most self-supervised methods for articulated object modeling reconstruct discrete interaction states and relate them via cross-state geometric consistency, yielding representational fragmentation and drift that hinder smooth control of articulated configurations. We introduce PD²GS, a novel framework that learns a shared canonical Gaussian field and models the arbitrary interaction state as its continuous deformation, jointly encoding geometry and kinematics. By associating each interaction state with a latent code and refining part boundaries using generic vision priors, PD²GS enables accurate and reliable part-level decoupling while enforcing mutual exclusivity between parts and preserving scene-level coherence. This unified formulation supports part-aware reconstruction, fine-grained continuous control, and accurate kinematic modeling, all without manual supervision. To assess realism and generalization, we release RS-Art, a real-to-sim RGB-D dataset aligned with reverse-engineered 3D models, supporting real-world evaluation. Extensive experiments demonstrate that PD²GS surpasses prior methods in geometric and kinematic accuracy, and in consistency under continuous control, both on synthetic and real data.

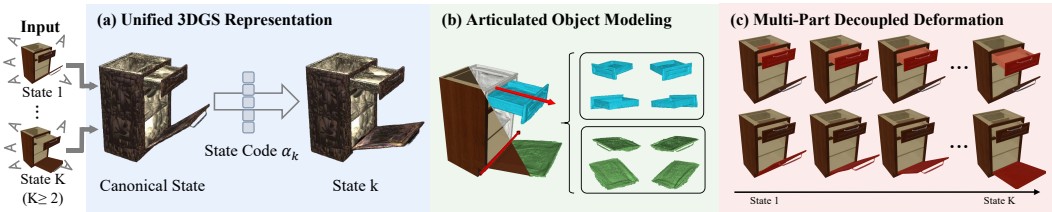

Figure 1: Given multi-view image sets at arbitrary interaction states, our framework (a) builds a unified 3D Gaussian representation for articulated objects, (b) achieves part-level articulated object reconstruction and joint motion analysis, and (c) enables continuous deformation of articulated objects with multi-part decoupling by interpolating latent codes.

## 1 Introduction

Articulated objects, from hinged doors and sliding drawers to foldable laptops, are ubiquitous in physical and virtual environments. Accurate 3D representations of articulated objects are critical for robotics (Simeonov et al., 2022; Si et al., 2024; Zhao et al., 2024), AR/VR (Yang et al., 2024a;

---

*Corresponding author.

Mangalam et al., 2024), and digital twins (Li et al., 2025; Guo et al., 2024). Prior work on articulated object modeling often builds on access to annotated 3D models with dense part and kinematic labels (Wang et al., 2019; Yan et al., 2019; Mu et al., 2021; Jiang et al., 2022; Wei et al., 2022; Wang et al., 2024). Such strong supervision, together with simplified appearance models and coarse geometric priors, limits applicability to objects with simple kinematic structure and low part diversity.

Recent advances in neural rendering, including NeRF (Mildenhall et al., 2021) and 3DGS (Kerbl et al., 2023), enable multi-view learning of appearance and geometry, offering a foundation for annotation-efficient articulated object modeling. PARIS (Liu et al., 2023) first employs a NeRF backbone (Mildenhall et al., 2021) for part-level modeling across two distinct interaction states. Subsequent studies (Guo et al., 2025; Wu et al., 2025), incorporating 3DGS (Kerbl et al., 2023), improve rendering fidelity and convergence speed. However, these methods typically assume a single movable part across two states, limiting them to simple single-joint objects. In response, several approaches infer multi-part decompositions by contrasting two state-specific 3D representations (Weng et al., 2024; Liu et al., 2025b), typically via explicit Marching-Cubes meshes and per-state Gaussian fields with subsequent joint alignment. This discretization confines optimization to pairwise geometric matching between the two states, typically augmented with physics-based alignment constraints, which relies on exact correspondences and precludes modeling continuous part motion. Additionally, operating on exactly two interaction states provides only sparse observations of the articulation, which limits estimation accuracy and hampers the resolution of motion ambiguities. Recent dynamic variants of NeRF (Mildenhall et al., 2021) and 3DGS (Kerbl et al., 2023) typically learn continuous representations of dynamic scenes conditioned on time, enabling high-fidelity novel-view synthesis from temporally ordered inputs (Pumarola et al., 2021; Wu et al., 2022; Yang et al., 2024c; Luiten et al., 2024). However, articulated object modeling explicitly requires the decoupling of motions at the part level, whereas existing dynamic rendering methods inherently capture only holistic scene transformations. Moreover, without continuous temporal data, these dynamic approaches struggle to effectively disentangle individual part motions from a limited number of discrete interaction states, often resulting in severe geometric distortions and blurry rendering artifacts.

Our key insight is that each interaction state of an articulated object can be modeled as a continuous deformation of a shared canonical 3D Gaussian field, with per-primitive deformation trajectories coherent within parts and distinct across parts, thereby inducing part semantics and enabling unsupervised part-level grouping. Building on this idea, we introduce **PD$^2$GS**, which realizes **P**art-Level **D**ecoupling and Continuous **D**eformation via **G**aussian **S**platting, enabling part-aware reconstruction and smooth transitions to previously unseen configurations (Fig. 1). We design Deformable Gaussian Splatting that represents each interaction state with a latent code used to parameterize per-primitive transformations of a shared canonical Gaussian field. To enable part-level decoupling of Gaussian primitives, we propose a coarse-to-fine segmentation procedure. We first obtain a coarse grouping by clustering primitives according to deformation-trajectory similarity with guidance from a vision–language model. We then refine part boundaries via a boundary-aware splitting stage that employs a tailored 3D-2D prompting scheme for SAM, yielding sharp interfaces and fine-grained per-primitive labels while preserving smooth part motion. Accordingly, our framework operates entirely within the 3DGS optimization paradigm, is agnostic to the number of interaction states, and avoids imposing geometric or physics-based constraints that are not expressible within the Gaussian primitive parameterization.

Moreover, most prior studies (Liu et al., 2023; Weng et al., 2024; Liu et al., 2025b; Wu et al., 2025) evaluate at most one instance per category on PartNet-Mobility (Xiang et al., 2020), a synthetic corpus with limited intra-category diversity and weak evidence for real-world generalization. We enlarge per-category coverage on PartNet-Mobility and additionally release **RS-Art**, a real-to-sim evaluation dataset that pairs multi-view RGB captures of real objects with their reverse-engineered 3D models, enabling more rigorous assessment of sim-to-real performance.

Our contributions can be summarized as follows:

- We introduce PD$^2$GS, a self-supervised framework that learns a canonical Gaussian field and realizes interaction states as its continuous deformations, enabling part-level decoupling and the joint recovery of geometry, appearance, and kinematics.
- We propose a coarse-to-fine segmentation of Gaussian primitives driven by deformation trajectories, with boundary refinement for sharp part interfaces and smooth part motion, yielding accurate part-aware segmentation.

- We evaluate multiple instances per category and release RS-Art, a real-to-sim evaluation dataset pairing real RGB captures with reverse-engineered 3D models. Comprehensive experiments demonstrate that PD$^2$GS outperforms prior methods.

## 2 RELATED WORK

**Articulated object modeling.** Early methods like PointNet require dense supervision and CAD-quality inputs (Qi et al., 2017a;b; Yan et al., 2019). Implicit SDFs relax input needs but demand watertight meshes, pre-aligned frames, and usually two interaction states (Mu et al., 2021; Wei et al., 2022; Jiang et al., 2022). Label-free NeRF and 3DGS approaches (Liu et al., 2023; Guo et al., 2025; Wu et al., 2025) support only single-joint, two-state objects. Recent two-field registration methods handle multiple parts but still assume known part counts and similar state configs (Weng et al., 2024; Liu et al., 2025b). Distinct from these reconstruction tasks, conditional generative approaches (Lei et al., 2023; Liu et al., 2025a) leverage diffusion models to synthesize articulated shapes from sparse inputs. However, these methods depend on extensive labeled datasets and often hallucinate plausible geometries that deviate from the true instance structure. PD$^2$GS overcomes these limitations with a latent-conditioned deformation field that supports arbitrary states and infers part structure from motion, enabling fully self-supervised multi-part articulation modeling.

**Dynamic Gaussian splatting.** Recent work extends static 3D Gaussians to dynamic scenes by parameterizing attributes as time-variant functions. Methods like Deformable 3DGS (Yang et al., 2024c; Luiten et al., 2024) employ time-conditioned MLPs to predict deformations, often enforcing physical priors. For efficient spacetime representation, approaches (Wu et al., 2024; Yang et al., 2024b) such as 4D-GS factorize the high-dimensional volume into compact feature planes to accelerate querying. Addressing the ambiguity of implicit deformations, recent works adopt explicit motion modeling via keyframes or neural flow fields for better tracking (Lee et al., 2024; Sun et al., 2025). In the context of articulated object modeling, these approaches inherently capture holistic scene dynamics, fundamentally lacking the ability to distinguish different motion semantics. Moreover, when applied to non-continuous interaction states, their reliance on temporal continuity leads to trajectory ambiguity and geometric artifacts. In contrast, PD$^2$GS employs a latent-conditioned deformation field to explicitly model deformations across discrete interaction states, establishing a unified representation that facilitates subsequent fine-grained part decoupling and control.

**SAM-based 3D segmentation.** Recent efforts lift SAM (Kirillov et al., 2023) to neural field segmentation by projecting 2D masks or distilling them into Gaussian features. SA3D (Cen et al., 2023) and SANeRF HQ (Liu et al., 2024) propagate and aggregate SAM masks in NeRF using multi-view and density cues. Within the Gaussian domain, SAGA (Cen et al., 2025) and SAGD (Hu et al., 2024) learn instance-aware embeddings or refine boundaries, while Gaussian Grouping (Ye et al., 2024) and SAM3D (Yang et al., 2023) directly lift 2D masks to 3D reconstructions. These approaches treat segmentation as static and depend on external 2D prompts or per-frame masks, failing to leverage motion cues across states. Our framework instead analyzes Gaussian motion trajectories to auto-generate state-aware prompts, then refines segments via SAM-guided splitting, enabling fully automatic part segmentation.

## 3 METHODOLOGY

This section explains the pipeline shown in Fig. 2. We first review the basics of 3D Gaussian Splatting (Sec. 3.1) and then attach a latent-conditioned deformation network that maps a canonical Gaussian field to every interaction state (Sec. 3.2). On the resulting state-specific fields, we detect rigid parts from motion cues (Sec. 3.3) and refine their boundaries with visibility-guided SAM prompting followed by boundary-aware Gaussian splitting (Sec. 3.4). The refined part-aware Gaussian field then supports the multi-task modeling of each articulated object (Sec. 3.5).

### 3.1 PRELIMINARIES: 3D GAUSSIAN SPLATTING

3D Gaussian Splatting (3DGS) (Kerbl et al., 2023) represents a scene by a finite set of oriented anisotropic Gaussian primitives

$$\mathcal{G} = \left\{ g_i = (\boldsymbol{\mu}_i,\ \boldsymbol{\Sigma}_i,\ \mathbf{c}_i,\ \alpha_i) \right\}_{i=1}^{N}, \tag{1}$$

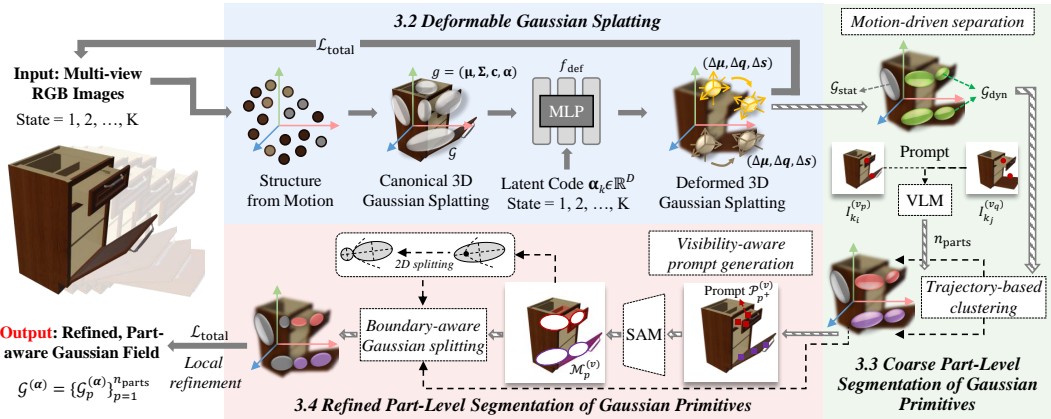

Figure 2: Overview of the PD$^2$GS pipeline. Solid arrows indicate differentiable modules that participate in the joint optimization, whereas dashed arrows correspond to non-differentiable post-processing stages executed outside the optimization loop.

where $\boldsymbol{\mu}_i \in \mathbb{R}^3$ denotes the center of the $i$-th Gaussian, $\boldsymbol{\Sigma}_i \in \mathbb{R}^{3\times3}$ its full covariance that encodes both scale and orientation, $\mathbf{c}_i \in [0,1]^3$ the RGB color, and $\alpha_i \in [0,1]$ the opacity.

A perspective camera is modeled by a projection $\Pi : \mathbb{R}^3 \to \mathbb{R}^2$. Linearization $\Pi$ at $\boldsymbol{\mu}_i$ maps the 3D covariance to a 2D screen-space covariance $\mathbf{S}_i \in \mathbb{R}^{2\times2}$ and the mean pixel coordinates $\mathbf{u}_i = \Pi(\boldsymbol{\mu}_i)$. Each primitive therefore induces the density

$$\rho_i(\mathbf{x}) = \alpha_i \exp\Big(-\tfrac{1}{2}(\mathbf{x} - \mathbf{u}_i)^\top \mathbf{S}_i^{-1}(\mathbf{x} - \mathbf{u}_i)\Big), \qquad \mathbf{x} \in \mathbb{R}^2. \tag{2}$$

With a depth ordering that proceeds from front to back, $\alpha$ compositing yields the pixel radiance

$$\mathbf{C}(\mathbf{x}) = \sum_{i=1}^{N} w_i(\mathbf{x})\,\mathbf{c}_i, \qquad w_i(\mathbf{x}) = \rho_i(\mathbf{x}) \prod_{j<i}\big[1 - \rho_j(\mathbf{x})\big]. \tag{3}$$

All parameters $\{\boldsymbol{\mu}_i, \boldsymbol{\Sigma}_i, \mathbf{c}_i, \alpha_i\}_{i=1}^{N}$ are optimized by minimizing a photometric loss between the rendered color $\mathbf{C}(\mathbf{x})$ and the corresponding ground-truth image.

## 3.2 DEFORMABLE GAUSSIAN SPLATTING

To capture the full continuum of interaction states we train an implicit network, conditioned on a latent code, that warps a canonical Gaussian field into state-specific configurations.

Each Gaussian primitive is $g_i = (\boldsymbol{\mu}_i, \boldsymbol{\Sigma}_i, \mathbf{c}_i, \alpha_i)$, with the covariance factorised as

$$\boldsymbol{\Sigma}_i \;=\; \mathbf{R}(\mathbf{q}_i)\,\mathrm{Diag}(\mathbf{s}_i)\,\mathbf{R}(\mathbf{q}_i)^\top, \tag{4}$$

where $\mathbf{R}(\boldsymbol{q}_i)$ converts the unit quaternion $\boldsymbol{q}_i \in SO(3)$ into its $3 \times 3$ rotation matrix, and $\mathrm{Diag}(\boldsymbol{s}_i)$ places the positive scale vector $\boldsymbol{s}_i \in \mathbb{R}^3_{>0}$ on the diagonal of a $3 \times 3$ matrix. This factorization separates orientation from per-axis stretch, keeping subsequent rotation and scale updates both intuitive and numerically stable.

Given a latent code $\boldsymbol{\alpha} \in \mathbb{R}^D$ that encodes one interaction state, a multi-layer perceptron $f_{\mathrm{def}}$ predicts per-primitive offsets

$$(\Delta\boldsymbol{\mu}_i, \Delta\mathbf{q}_i, \Delta\mathbf{s}_i) \;=\; f_{\mathrm{def}}(\boldsymbol{\mu}_i, \mathbf{q}_i, \mathbf{s}_i \mid \boldsymbol{\alpha}), \tag{5}$$

with $\Delta\boldsymbol{\mu}_i \in \mathbb{R}^3$, $\Delta\boldsymbol{s}_i \in \mathbb{R}^3$, and an unconstrained four-vector $\Delta\boldsymbol{q}_i$ that is normalized to $SO(3)$ by $\Delta\boldsymbol{q}_i \leftarrow \Delta\boldsymbol{q}_i / \|\Delta\boldsymbol{q}_i\|_2$.

Applying the learned offsets converts the canonical Gaussian field into the configuration of a particular interaction state $k$:

$$\boldsymbol{\mu}_i^{(k)} = \boldsymbol{\mu}_i + \Delta\boldsymbol{\mu}_i, \qquad \mathbf{s}_i^{(k)} = \mathbf{s}_i + \Delta\mathbf{s}_i, \qquad \mathbf{q}_i^{(k)} = \Delta\mathbf{q}_i \otimes \mathbf{q}_i, \tag{6}$$

where $\otimes$ denotes multiplication of quaternions. Normalizing $\Delta \boldsymbol{q}_i$ produces a unit quaternion, so $\boldsymbol{q}_i^{(k)}$ remains a valid orientation. Eq. 6 therefore converts the canonical Gaussians $\{\boldsymbol{\mu}_i, \boldsymbol{s}_i, \boldsymbol{q}_i\}$ into the state-specific set $\{\boldsymbol{\mu}_i^{(k)}, \boldsymbol{s}_i^{(k)}, \boldsymbol{q}_i^{(k)}\}$ that describes the interaction state $k$.

For the $K$ observed interaction states of the object, we jointly optimize the network parameters $\theta_{f_{\text{def}}}$, the latents $\{\boldsymbol{\alpha}_k\}_{k=1}^{K}$, and the canonical Gaussian parameters by minimizing the following total loss:

$$\mathcal{L}_{\text{total}} = \underbrace{\sum_{k,v} \sum_{\mathbf{x} \in \Omega_k^{(v)}} \|\mathbf{C}_k^{(v)}(\mathbf{x}) - \mathbf{I}_k^{(v)}(\mathbf{x})\|_1}_{\mathcal{L}_{\text{photo}}} + \mathcal{L}_{\text{D}_{\text{SIMM}}}, \tag{7}$$

where $\mathbf{C}_k^{(v)}$ is the color rendered in pixel $\mathbf{x}$, $\mathbf{I}_k^{(v)}$ is the corresponding ground truth image, and $\mathcal{L}_{\text{D}_{\text{SIMM}}}$ is the canonical 3DGS density similarity term that discourages overlapping Gaussians with dissimilar colors.

After convergence, every latent $\boldsymbol{\alpha}_k$ produces a state-specific geometry $\{\boldsymbol{\mu}_i^{(k)}, \boldsymbol{s}_i^{(k)}, \boldsymbol{q}_i^{(k)}\}_{i=1}^{N}$ while sharing the appearance attributes $\{\mathbf{c}_i, \alpha_i\}_{i=1}^{N}$. A single model therefore yields a coherent Gaussian scene for each interaction state and jointly encodes geometry, appearance, and articulation.

### 3.3 COARSE PART-LEVEL SEGMENTATION OF GAUSSIAN PRIMITIVES

**Motion-driven separation of static and dynamic Gaussians.** Without object masks or semantic priors, we detect dynamic primitives by computing the maximum Euclidean displacement of each Gaussian center across the $K$ interaction states. Let $\boldsymbol{\mu}_i^{(k)} \in \mathbb{R}^3$ be the center of Gaussian primitives $g_i$ in state $k$. We define its maximal displacement as

$$d_i = \max_{j,k \in \{1,\ldots,K\}} \|\boldsymbol{\mu}_i^{(j)} - \boldsymbol{\mu}_i^{(k)}\|_2, \quad \hat{d}_i = d_i / \max_r d_r \in [0,1]. \tag{8}$$

The Gaussians with $\hat{d}_i \geq \tau_{\text{mot}}$ form the dynamic set $\mathcal{G}_{\text{dyn}}$, and the remainder $\mathcal{G}_{\text{stat}}$ is static, where the threshold $\tau_{\text{mot}} \in [0,1]$ specifies the fraction of the maximum displacement of the entire scene that qualifies as motion.

**Estimating the number of motion parts via a VLM.** To infer how many rigid parts appear in $\mathcal{G}_{\text{dyn}}$, we sample $M$ image pairs $(I_{k_i}^{(v_p)}, I_{k_j}^{(v_q)})$ with different states $k_i \neq k_j$. A visual language model (VLM) (Dai et al., 2023) is queried with the fixed prompt:

> *"Compare the two images. How many components moved?"*
> *"Answer: 'Number of moved components: [N]'."*

The number of parts $n_{\text{parts}}$ is set in the mode of the VLM predictions $M$, i.e., the integer that occurs most frequently, making the estimate insensitive to occasional miscounts.

**Trajectory-based clustering of dynamic Gaussians.** For each $g_i \in \mathcal{G}_{\text{dyn}}$ we assemble a motion descriptor

$$\mathbf{f}_i = \left[\hat{\Delta}\boldsymbol{\mu}_i^{(1)}, \|\Delta\boldsymbol{\mu}_i^{(1)}\|_2, \ldots, \hat{\Delta}\boldsymbol{\mu}_i^{(K-1)}, \|\Delta\boldsymbol{\mu}_i^{(K-1)}\|_2\right] \in \mathbb{R}^{4(K-1)}, \tag{9}$$

where

$$\Delta\boldsymbol{\mu}_i^{(k)} = \boldsymbol{\mu}_i^{(k+1)} - \boldsymbol{\mu}_i^{(k)}, \qquad \hat{\Delta}\boldsymbol{\mu}_i^{(k)} = \frac{\Delta\boldsymbol{\mu}_i^{(k)}}{\|\Delta\boldsymbol{\mu}_i^{(k)}\|_2 + \varepsilon}. \tag{10}$$

Unit directions encode orientation and lengths keep relative travel, so the descriptor is translation/rotation aware yet scale balanced. After normalization of $\ell_2$, we cluster the descriptors with K-means (Arthur & Vassilvitskii, 2006) using $K = n_{\text{parts}}$. On the unit sphere, Euclidean distance equals cosine distance, therefore Gaussians that share one rigid motion, even at different amplitudes, are grouped together.

Clusters containing fewer than 2% of dynamic Gaussians are regarded noise and merged into the nearest larger cluster. Each dynamic primitive then receives a part label $c_i \in \{1, \ldots, n_{\text{parts}}\}$, while static ones are labeled 0. These coarse labels initialize the fine-grained stage that follows.

## 3.4 REFINED PART-LEVEL SEGMENTATION OF GAUSSIAN PRIMITIVES

**Visibility-aware prompt generation for SAM.** To sharpen the coarse part labels in Sec. 3.3 we first render each part through a visibility filter and then draw sparse point prompts for the Segment Anything Model (SAM) (Kirillov et al., 2023).

- **Per-Gaussian contribution at a pixel.** For a pixel $\mathbf{x}$ in view $v$, the contribution weight $w_i^{(v)}(\mathbf{x})$ from Gaussian $g_i = (\boldsymbol{\mu}_i, \boldsymbol{\Sigma}_i, \mathbf{c}_i, \alpha_i)$ is obtained by first evaluating its screen-space density $\rho_i^{(v)}(\mathbf{x})$ (Eq. 2) and then applying the front-to-back compositing rule (Eq. 3), which together map every three-dimensional primitive to a continuous weight field whose value at $\mathbf{x}$ equals $w_i^{(v)}(\mathbf{x})$.

- **Part specific visibility confidence.** For each pixel we accumulate the weight of part $p$ as $w_p^{(v)}(\mathbf{x}) = \sum_{g_i \in \mathcal{G}_p} w_i^{(v)}(\mathbf{x})$ and compare it with the strongest competitor $w_{\max, \neg p}^{(v)}(\mathbf{x}) = \max_{q \neq p} w_q^{(v)}(\mathbf{x})$. The pixel is assigned to part $p$ when

$$w_p^{(v)}(\mathbf{x})/\{w_{\max, \neg p}^{(v)}(\mathbf{x}) + \varepsilon\} > \tau_{\mathrm{vis}}, \tag{11}$$

where $\tau_{\mathrm{vis}}$ is a visibility threshold, tuned on one or two representative instances, and $\varepsilon$ avoids division by zero. The winner margin test keeps only pixels in which part $p$ clearly dominates, thus reducing the ambiguity at overlaps.

- **Sampling prompts.** We define the set of visible pixels corresponding to part $p$ in view $v$ as $\Omega_{p^+}^{(v)} = \{\mathbf{x} \mid \text{pixel assigned to part } p\}$ and the non-contributing set as $\Omega_{p^-}^{(v)} = \{\mathbf{x} \mid w_p^{(v)}(\mathbf{x}) = 0\}$. Furthest point sampling (FPS) on $\Omega_{p^+}^{(v)}$ and $\Omega_{p^-}^{(v)}$ selects ten positive prompts $\mathcal{P}_{p^+}^{(v)}$ and twenty negative prompts $\mathcal{P}_{p^-}^{(v)}$, which together with the RGB image are fed to SAM to obtain the mask $\mathcal{M}_p^{(v)}$.

**Boundary-aware Gaussian splitting.** For each Gaussian $g_i$ we find its nearest view $v_{\mathrm{near}}$ by ray sampling and project the ellipsoid to a two-dimensional ellipse. If the ellipse major axis extends beyond the mask of the part $\mathcal{M}_p^{(v_{\mathrm{near}})}$, the Gaussian becomes a boundary candidate.

We follow SAGD (Hu et al., 2024) to split each candidate, as shown in the bottom part of Fig. 2. The fraction of the major axis that lies inside the mask is measured in image space and the same ratio is applied to the center $\boldsymbol{\mu}_i$ and the scale vector $\mathbf{s}_i$, resulting in two children $g_i^{(\mathrm{part})}$ and $g_i^{(\mathrm{bg})}$. The part child retains the in-mask portion and the background child captures the overflow. The background child $g_i^{(\mathrm{bg})}$ is re-evaluated in its nearest view; if its ellipse still crosses another part mask the split is applied recursively until every descendant lies inside exactly one mask.

**Part-aware Gaussian field via local refinement.** Finally, parameters of unsplit Gaussians remain fixed, whereas all new children are locally fine-tuned for a few iterations on $\mathcal{L}_{\mathrm{total}}$ (updating $\mathbf{q}$ and $\alpha$ only) to restore the photo consistency. The recursive split-and-tune process produces sharply aligned, mask-consistent part boundaries while leaving the converged Gaussian field undisturbed.

The entire procedure yields a refined, part-aware Gaussian field with interaction state

$$\mathcal{G}^{(\boldsymbol{\alpha})} = \{\mathcal{G}_p^{(\boldsymbol{\alpha})}\}_{p=1}^{n_{\mathrm{parts}}}, \qquad \mathcal{G}_p^{(\boldsymbol{\alpha})} = \{(\boldsymbol{\mu}_i^{(\boldsymbol{\alpha})}, \boldsymbol{\Sigma}_i^{(\boldsymbol{\alpha})}, \mathbf{c}_i, \alpha_i)\}_{i \in \mathcal{G}_p}, \tag{12}$$

where $\boldsymbol{\alpha} \in \mathbb{A}$ indexes the interaction state ($|\mathbb{A}| = K$) and $\mathcal{G}_p$ denotes the set of Gaussian indices belonging to part $p$. Each state-specific field $\mathcal{G}^{(\boldsymbol{\alpha})}$ encodes a coherent multi-part configuration, while the union over all $\boldsymbol{\alpha}$ compactly describes the full articulated motion range of the object.

## 3.5 MULTI-TASK MODELING FOR ARTICULATED OBJECTS

Given the part-aware Gaussian field $\mathcal{G}^{(\boldsymbol{\alpha})} = \{\mathcal{G}_p^{(\boldsymbol{\alpha})}\}_{p=1}^{n_{\mathrm{parts}}}$, we extract three part-level outputs. First, every subset $\mathcal{G}_p^{(\boldsymbol{\alpha})}$ is rendered in depth and color images in all calibrated views; marching cubes applied to the resulting volume produce the mesh $\mathcal{M}_p \in \mathbb{R}^{|V_p| \times 3}$. Second, to classify the type of joint part $p$, we align two interaction states with the Kabsch algorithm (Lawrence et al., 2019) and inspect the residual displacement: a low-rank residual indicates a *revolute* joint, while a full-rank

residual indicates a *prismatic* joint. Third, the trajectories of the Gaussian centroids belonging to part $p$ are fitted with a minimal motion model. A revolute joint is parameterized by a pivot point $\mathbf{p} \in \mathbb{R}^3$ and a unit quaternion $\mathbf{q} \in \mathbb{R}^4$ with $\|\mathbf{q}\| = 1$. A prismatic joint is parameterized by a unit slide axis $\mathbf{a} \in \mathbb{R}^3$ with $\|\mathbf{a}\| = 1$ and a translation distance $d \in \mathbb{R}$. More details are provided in Sec. A of the Appendix.

## 4 THE RS-ART DATASET

We establish **RS-Art**, a high-quality, multi-modal benchmark that bridges the gap between synthetic and real-scene evaluation for articulated-object modeling. It comprises real-world captures of articulated objects alongside their reverse-engineered, part-level digital counterparts, providing a rare combination of dense sensory data and precise structural ground truth. Covering six representative object categories (drawers, desk lamps, eyeglasses, floppy-disk drives, woven baskets, and phone/laptop stands), the dataset includes three diverse instances per category. Each object is recorded in seven distinct articulation states, spanning both compound multi-joint configurations and single-joint extrema, resulting in over 400 wide-baseline RGB-D observations per instance. For every object, we reconstruct a high-fidelity, textured, multi-part mesh annotated with joint axes and motion limits. All assets are released in URDF format, enabling direct deployment in physics-based simulators for downstream tasks and rigorous, physically grounded evaluation. Further details of the dataset are provided in Sec. B of the Appendix.

## 5 EXPERIMENTS

### 5.1 EXPERIMENTAL SETTING

**Datasets.** Most studies evaluate only one object per category, which limits the statistical power. We extend PartNet-Mobility (Xiang et al., 2020) by selecting eight categories and two to three instances per category, each with two movable parts, so all methods face the same kinematic complexity. We also include four objects with three movable parts to test the robustness under higher joint counts. Real-scene performance is measured on the RS-Art we proposed. Together, the enlarged PartNet-Mobility split and RS-Art form a well-balanced platform for both synthetic and real-world scenarios.

**Baselines.** We compare our method with recent SOTA methods. The *single-joint* baselines include **PARIS** (Liu et al., 2023) and **ArticulatedGS** (Guo et al., 2025). Each single-joint pipeline was run once for every joint: in each run, the input images show motion of a single joint while the other parts remain fixed, and the per-joint outputs were merged for evaluation. The *multi-joint* baselines include **DTArt** (Weng et al., 2024) and **ArtGS** (Liu et al., 2025b). Both baselines require a prespecified part count, whereas our method infers the parts automatically. For the dataset setup, two-state baselines sample 100 views per state (200 RGB images; DTArt also requires depth). Matching this budget, we sample 50 views across four states, totaling 200 RGB inputs. For more information on our method restricted to two interaction states, refer to Sec. D.1 of the Appendix.

**Metrics.** We adopt exactly the evaluation metrics defined in DTArt (Weng et al., 2024), covering the geometry reconstruction and precision of the joint parameter. CD-w, CD-s, and CD-m are used to measure the Chamfer Distance for the whole object, static parts, and movable parts in mesh reconstruction. Axis Ang, Axis Pos, and Part Motion are used to evaluate the errors between predicted and ground-truth joint axes parameters. The only extension is that our reported scores are aggregated over all movable parts of an object, rather than being limited to a single target part.

Additional implementation details are in Sec. C of the Appendix.

### 5.2 COMPARISON ON OBJECTS WITH MULTIPLE MOVABLE PARTS

Tab. 1 reports quantitative results on our expanded PartNet-Mobility split, which contains objects with *two* independently actuated parts. Previous studies usually average across several runs, yet we observe large run-to-run variance in several baselines, to reveal this sensitivity, we list the *first* run for every method. These numbers are not cherry-picked, and subsequent runs often produce even lower scores. PD$^2$GS needs no manual part specification and still outperforms the baselines in most metrics. Fig. 3, which also shows an object with three movable parts, qualitatively confirms that our

Table 1: Results on the PartNet-Mobility Dataset. A superscript * denotes that the method failed to obtain a valid result for at least one object in the corresponding category. Note that DTArt requires additional *depth* input, while other methods rely on RGB inputs.

| Metric | Method | Box | Door | Eyeglasses | Faucet | Oven | Fridge | Storage | Table | Mean |
|---|---|---|---|---|---|---|---|---|---|---|
| | | | | | Synthetic Objects | | | | | |
| Axis ↓ Ang 0 | PARIS (Liu et al., 2023) | 36.27 | 24.83 | 64.35* | 13.75* | 54.88 | 69.12 | 27.05 | 15.60 | 38.23 |
| | ArticulatedGS (Guo et al., 2025) | 0.98 | 1.43 | 2.25 | 5.14 | 4.13 | 14.14 | 23.24 | 1.35 | 6.58 |
| | DTArt (Weng et al., 2024) | 20.65 | 0.38 | 18.18 | 1.06 | 3.37 | 40.99 | 0.46 | 0.40 | 10.69 |
| | ArtGS (Liu et al., 2025b) | 0.01 | 0.35 | 0.06 | 8.05 | 2.33* | 0.13* | 6.05 | 1.98 | 2.37 |
| | PD²GS (Ours) | **0.49** | **0.31** | **0.06** | **0.88** | **0.34** | **0.09** | **0.21** | **0.25** | **0.33** |
| Axis ↓ Ang 1 | PARIS (Liu et al., 2023) | 2.89 | 31.69 | 3.25* | 2.65* | 17.29 | 49.39 | 18.00 | 48.61 | 21.72 |
| | ArticulatedGS (Guo et al., 2025) | 5.12 | 0.77 | 2.12 | 4.12 | 5.25 | 2.13 | 0.12 | 0.52 | 2.52 |
| | DTArt (Weng et al., 2024) | 42.60 | 0.75 | 1.62 | 1.44 | 14.21 | 2.17 | 0.22 | 18.12 | 10.14 |
| | ArtGS (Liu et al., 2025b) | 1.36 | 0.68 | 0.08 | 17.02 | 9.24* | 1.41* | 0.01 | 7.66 | 4.68 |
| | PD²GS (Ours) | **0.28** | **0.49** | **1.22** | **0.58** | **0.12** | **0.73** | **0.04** | **0.28** | **0.47** |
| Axis ↓ Pos 0 | PARIS (Liu et al., 2023) | 0.14 | 0.50 | 1.12* | 0.99* | 0.29 | 0.13 | 0.10 | 0.29 | 0.45 |
| | ArticulatedGS (Guo et al., 2025) | 0.97 | 0.26 | 2.11 | 0.24 | 3.46 | 0.26 | 0.74 | 0.79 | 1.10 |
| | DTArt (Weng et al., 2024) | 7.09 | 0.01 | 4.96 | 0.02 | 14.89 | 11.35 | 0.07 | 0.03 | 4.80 |
| | ArtGS (Liu et al., 2025b) | 0.06 | 0.03 | 0.06 | 0.01 | 1.9* | 0.02* | 1.28 | 39.14 | 5.31 |
| | PD²GS (Ours) | **0.06** | **0.02** | **0.02** | **0.04** | **0.27** | **0.16** | **0.03** | **0.01** | **0.08** |
| Axis ↓ Pos 1 | PARIS (Liu et al., 2023) | 0.33 | 0.40 | 0.77* | 0.44* | 0.11 | 0.18 | 0.20 | 0.02 | 0.31 |
| | ArticulatedGS (Guo et al., 2025) | 0.24 | 0.62 | 0.24 | 1.24 | 2.15 | 0.15 | 0.24 | 0.65 | 0.69 |
| | DTArt (Weng et al., 2024) | 3.96 | 0.21 | 11.75 | 0.02 | 32.87 | 0.15 | 0.01 | 0.01 | 6.12 |
| | ArtGS (Liu et al., 2025b) | 0.51 | 0.03 | 0.04 | 0.05 | 6.24* | 0.04* | 0.01 | 5.98 | 1.61 |
| | PD²GS (Ours) | **0.00** | **0.02** | **0.09** | **0.04** | **0.08** | **0.03** | **0.00** | **0.01** | **0.03** |
| Part ↓ Motion 0 | PARIS (Liu et al., 2023) | 93.80 | 151.50 | 97.24* | 65.79* | 59.28 | 99.33 | 45.24 | 16.38 | 78.57 |
| | ArticulatedGS (Guo et al., 2025) | 4.24 | 2.65 | 34.24 | 2.46 | 6.35 | 3.35 | 2.33 | 0.36 | 7.00 |
| | DTArt (Weng et al., 2024) | 42.85 | 0.35 | 13.91 | 0.07 | 70.02 | 36.73 | 0.07 | 0.55 | 20.57 |
| | ArtGS (Liu et al., 2025b) | 2.78 | 0.05 | 0.07 | 6.80 | 1.67* | 0.03* | 6.69 | 8.97 | 3.38 |
| | PD²GS (Ours) | **2.09** | **0.05** | **0.02** | **0.95** | **0.74** | **0.10** | **0.43** | **0.32** | **0.59** |
| Part ↓ Motion 1 | PARIS (Liu et al., 2023) | 47.89 | 33.11 | 87.44* | 19.35* | 28.29 | 150.42 | 57.09 | 81.32 | 63.11 |
| | ArticulatedGS (Guo et al., 2025) | 20.04 | 7.24 | 0.63 | 9.24 | 2.58 | 19.87 | 0.90 | 0.23 | 7.59 |
| | DTArt (Weng et al., 2024) | 63.48 | 89.31 | 104.92 | 1.44 | 65.79 | 4.91 | 0.05 | 0.21 | 41.26 |
| | ArtGS (Liu et al., 2025b) | 15.68 | 0.05 | 0.08 | 16.89 | 3.87* | 0.03* | 0.16 | 24.09 | 7.60 |
| | PD²GS (Ours) | **0.87** | **0.05** | **0.02** | **0.50** | **0.34** | **0.02** | **0.21** | **0.06** | **0.26** |
| CD-s ↓ | PARIS (Liu et al., 2023) | 6.93 | 17.90 | 23.35* | 9.36* | 13.45 | 18.96 | 10.44 | 3.45 | 12.98 |
| | ArticulatedGS (Guo et al., 2025) | 18.17 | 167.53 | 15.14 | 0.97 | 29.43 | 23.80 | 16.18 | 5.92 | 34.64 |
| | DTArt (Weng et al., 2024) | 2.45 | 20.60 | 49.28 | 0.49 | 2.60 | 4.06 | 1.27 | 8.65 | 11.17 |
| | ArtGS (Liu et al., 2025b) | 3.48 | 0.35 | 0.13 | 1.05 | 5.44* | 9.69* | 5.10 | 9.04 | 4.29 |
| | PD²GS (Ours) | **1.97** | **0.80** | **0.11** | **0.64** | **1.66** | **4.15** | **2.78** | **4.27** | **2.05** |
| CD-m 0 ↓ | PARIS (Liu et al., 2023) | 82.05 | 42.22 | 86.37* | 34.79* | 107.25 | 212.08 | 182.85 | 16.58 | 95.52 |
| | ArticulatedGS (Guo et al., 2025) | 9.65 | 0.84 | 26.88 | 35.63 | 1.91 | 11.24 | 7.32 | 7.24 | 12.59 |
| | DTArt (Weng et al., 2024) | 363.75 | 0.22 | 47.24 | 0.22 | 157.26 | 0.63 | 0.23 | 46.30 | 76.98 |
| | ArtGS (Liu et al., 2025b) | 2.47 | 0.44 | 0.16 | 0.38 | 8.89* | 0.86* | 3.46 | 143.78 | 20.06 |
| | PD²GS (Ours) | **2.44** | **0.15** | **0.09** | **0.71** | **1.79** | **0.82** | **0.03** | **5.40** | **1.43** |
| CD-m 1 ↓ | PARIS (Liu et al., 2023) | 128.58 | 32.08 | 98.54* | 48.86* | 301.18 | 76.82 | 125.06 | 76.18 | 110.91 |
| | ArticulatedGS (Guo et al., 2025) | 8.78 | 0.44 | 13.47 | 0.27 | 10.92 | 1.15 | 179.27 | 12.82 | 28.39 |
| | DTArt (Weng et al., 2024) | 198.70 | 1.17 | 1133.05 | 0.22 | 140.52 | 0.36 | 0.27 | 0.22 | 184.31 |
| | ArtGS (Liu et al., 2025b) | 3.35 | 0.42 | 0.17 | 0.46 | 8.47* | 0.86* | 2.25 | 1145.91 | 145.23 |
| | PD²GS (Ours) | **1.84** | **0.43** | **0.13** | **0.74** | **2.15** | **0.27** | **0.11** | **0.76** | **0.80** |
| CD-w ↓ | PARIS (Liu et al., 2023) | 6.30 | 12.59 | 17.24* | 9.98* | 10.93 | 18.89 | 8.86 | 2.81 | 10.95 |
| | ArticulatedGS (Guo et al., 2025) | 13.75 | 12.59 | 19.03 | 2.78 | 23.22 | 18.89 | 50.59 | 5.81 | 18.33 |
| | DTArt (Weng et al., 2024) | 1.08 | 1.40 | 0.15 | 0.50 | 1.73 | 0.98 | 10.17 | 1.39 | 2.18 |
| | ArtGS (Liu et al., 2025b) | 5.91 | 1.56 | 0.17 | 0.58 | 7.35* | 16.82* | 8.66 | 12.73 | 6.72 |
| | PD²GS (Ours) | **4.07** | **1.24** | **0.13** | **0.47** | **1.48** | **10.16** | **5.48** | **1.14** | **3.02** |

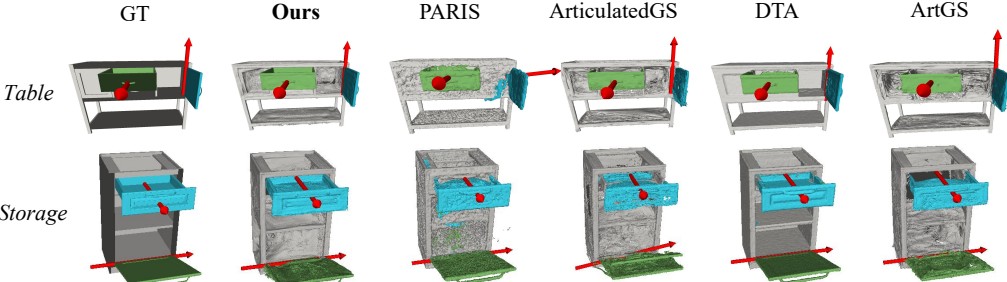

Figure 3: Qualitative multi-task modeling results on multi-part articulated objects.

joint estimates align more closely with the ground truth. DTArt (Weng et al., 2024) benefits from depth input and therefore renders smoother surfaces, but still lags behind in articulation accuracy. Overall, the use of a latent-conditioned deformation field spanning multiple interaction states yields

markedly higher stability than single-state or pairwise pipelines. Results on more objects with *three* movable parts are presented in Sec. D.2 of the Appendix, where the same trend persists.

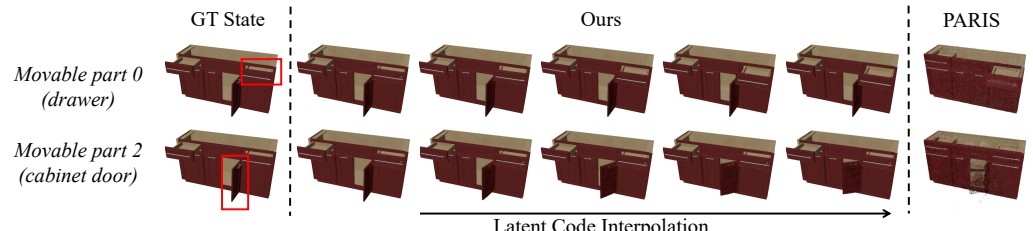

Figure 4: Generalization to unseen interaction states. We interpolate the latent code while deforming only the Gaussians that belong to a chosen part.

## 5.3 GENERALIZATION TO UNSEEN INTERACTION STATES

Fig. 4 and Fig. 5 illustrate that our method is the *only* one that is capable of disentangling and controlling the motion of several parts independently while maintaining plausible geometry in interaction states never shown during training. Fig. 5 specifically demonstrates how we perform interpolation between the start state and the end state. By interpolating the latent code and deforming only the Gaussians that belong to a chosen part, we generate smooth, collision-free trajectories for every drawer and for the door. In contrast, PARIS (Liu et al., 2023) cannot keep the parts separated, often leading to part overlap or blending due to the inherent multi-solution nature of its method. The capac-

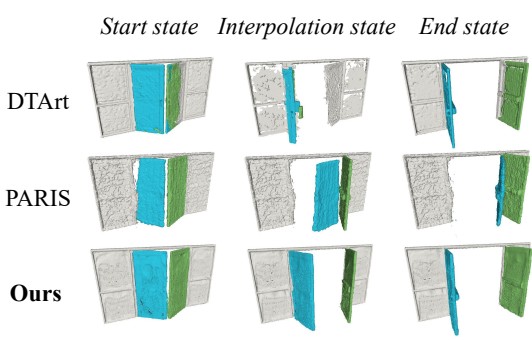

Figure 5: Qualitative reconstruction results on intermediate states.

ity to vary multiple joints independently and to synthesize genuinely novel poses represents a substantial step toward high-fidelity digital twin modeling of articulated objects.

Table 2: Quantitative comparison with deformable NeRF- and 3DGS-based methods on unseen-state rendering.

| Method | PSNR↑ | SSIM↑ | LPIPS↓ |
|---|---|---|---|
| D-NeRF (Pumarola et al., 2021) | 18.38 | 0.86 | 0.19 |
| D$^2$NeRF (Wu et al., 2022) | 21.34 | 0.85 | 0.16 |
| Dynamic3DGS (Katsumata et al., 2023) | 25.96 | 0.89 | 0.17 |
| 4D-GS (Wu et al., 2024) | 23.61 | 0.92 | 0.13 |
| ArtGS (Liu et al., 2025b) | 26.82 | 0.92 | 0.12 |
| PD$^2$GS (Ours) | **28.02** | **0.93** | **0.09** |

To further substantiate this capability, we extend our analysis by comparing our method with recent dynamic scene modeling approaches (Pumarola et al., 2021; Wu et al., 2022; Yang et al., 2024c; Luiten et al., 2024), including deformable NeRF- and 3DGS-based method. Following the protocol in Tab. 1, we evaluate on the Storage category, supplying each method with four discrete interaction states as input. We assess the capability of each method to synthesize intermediate interaction states by rendering novel views interpolated between input configurations. Additionally, for the two-state-based ArtGS (Liu et al., 2025b), we apply its learned joint parameters to rigidly transform the reconstructed Gaussian field to generate the corresponding interpolated poses. As shown in Fig. 6 and Tab. 2, our method achieves significantly superior performance in both visual fidelity and quantitative metrics. Notably, it effectively eliminates geometric artifacts that commonly plague holistic deformation methods.

## 5.4 RESULT ON THE RS-ART DATASET

Fig. 7 shows qualitative results on a real object from our RS-Art dataset. Despite sensor noise and challenging lighting, the method reconstructs detailed part-level geometry and texture and reliably extrapolates to interaction states not seen during training. These observations indicate that our method remains stable on real data. Further results are provided in Sec. D.3 of the Appendix.

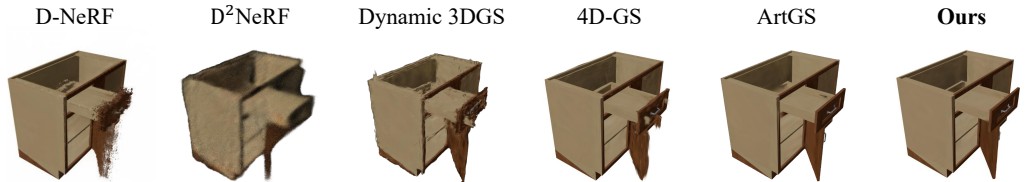

Figure 6: Qualitative comparison with deformable NeRF- and 3DGS-based methods on unseen-state rendering.

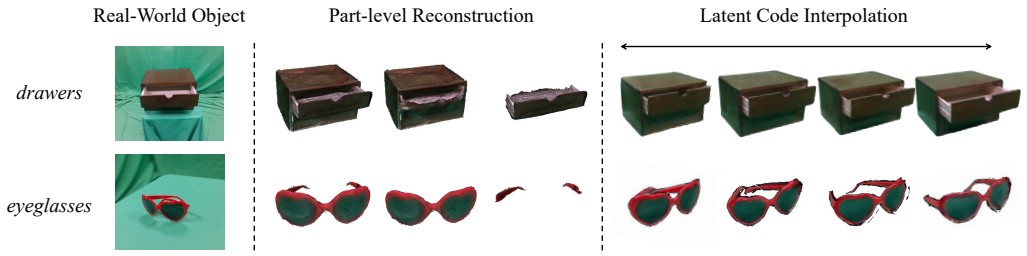

Figure 7: Real object validation on RS-Art.

## 5.5 ABLATION STUDIES

To evaluate the effectiveness of our coarse-to-fine strategy (Sec. 3.3 and 3.4), we conduct an ablation study by removing the refinement stage. Fig. 8 compares the results *without* the refinement stage (top) to those *with* it (bottom) on an oven containing multiple movable panels. For the variant *without* refinement, we reconstruct multiple parts directly from the K-means clusters obtained in the coarse stage. Without refinement, the reconstructed parts bleed into one another and exhibit artifacts; after refinement, the mesh shows clean separation, sharper boundaries, and no interpenetration. The improvement stems from visibility-guided SAM

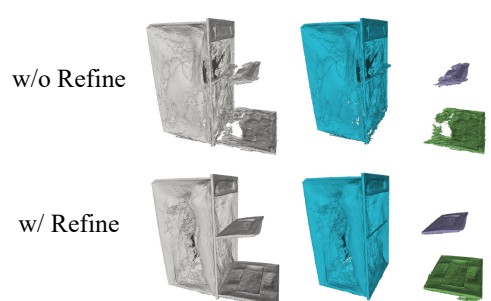

Figure 8: Ablation on the coarse–to–fine refinement stage.

prompting and boundary-aware Gaussian splitting, which force the Gaussian field to conform to the per-part masks. We provide additional visual and quantitative comparisons, along with further ablation analysis on movable-component count estimation via VLM, SAM-based segmentation, and each pipeline module, in Sec. D.7 of the Appendix.

## 6 CONCLUSION

We introduce PD$^2$GS, the fully self-supervised framework for articulated object modeling within a unified 3D Gaussian Splatting paradigm. A latent-conditioned deformation network warps a canonical Gaussian field across interaction states, and a visibility-guided coarse-to-fine scheme refines deformation trajectories into part-aware Gaussian clusters. The resulting field supports part-level mesh extraction, joint typing, and motion parameter estimation. In an expanded PartNet-Mobility split and the novel RS-Art dataset, PD$^2$GS surpasses existing methods without requiring part count in advance, and its learned deformation space generalizes smoothly to unseen interaction states, a property well suited to digital twin scenarios. The present implementation assumes accurate camera poses and cannot reconstruct regions that remain fully occluded in all views, leaving robustness to pose noise and reasoning about unseen structures as directions for future work.

ACKNOWLEDGEMENTS

This research was supported in part by the Natural Science Foundation of Anhui under Grant No. 2508085QF241, in part by the National Natural Science Foundation of China under Grant No. 62571055 and No. 62431015.

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

# Appendix

The supplementary material is organized as follows. Sec. A elaborates on the technical details; Sec. B details the acquisition and reverse modeling pipeline used to construct the RS-Art dataset; Sec. C lists the complete experimental settings; Sec. D presents additional quantitative and qualitative results; and Sec. E discusses the current limitations of our approach and directions for future work. All code and datasets will be released upon publication.

## A METHOD DETAILS

### A.1 BOUNDARY-AWARE GAUSSIAN SPLITTING

**Selecting boundary candidates.** For each 3D Gaussian primitive $g_i = (\boldsymbol{\mu}_i, \boldsymbol{\Sigma}_i, \mathbf{c}_i, \alpha_i)$, we first locate its ray–sampling nearest view $v_{\text{near}}$. Assuming a local affine projection, the covariance $\boldsymbol{\Sigma}_i$ is mapped to the image plane by

$$\boldsymbol{\Sigma}_i' = \mathbf{J}\mathbf{W}\boldsymbol{\Sigma}_i\mathbf{W}^\top\mathbf{J}^\top, \tag{13}$$

where $\mathbf{W}$ and $\mathbf{J}$ denote the camera projection matrix and its Jacobian, respectively (Kerbl et al., 2023). The resulting $2 \times 2$ matrix $\boldsymbol{\Sigma}_i'$ defines an ellipse; its major axis is aligned with the eigenvector of $\boldsymbol{\Sigma}_i'$ that has the largest eigenvalue. If either end of this axis lies outside the part mask $\mathcal{M}_p^{(v_{\text{near}})}$, the primitive is marked as *boundary candidate*.

**Geometric split ratio.** Let $A^*$ and $B^*$ be the two end points of the projected major axis and let $O^*$ be their intersection with the mask boundary ($A^*$ lies inside, $B^*$ outside). Following the split strategy of Hu *et al.* (Hu et al., 2024), the in-mask proportion in image space is

$$\lambda_{\text{2D}} = \frac{\|O^*A^*\|}{\|A^*B^*\|}. \tag{14}$$

Because the projection is affine locally, the same ratio applies in 3D. With $\mathbf{e}$ denoting the unit eigenvector of $\boldsymbol{\Sigma}_i$ that corresponds to its largest eigenvalue and $s_i$ the length of the associated axis, the two children are

- *Part child:* $\boldsymbol{\mu}_i^{\text{part}} = \boldsymbol{\mu}_i + \frac{1-\lambda_{\text{2D}}}{2} s_i \mathbf{e}$, $s_i^{\text{part}} = \lambda_{\text{2D}} s_i$;
- *Background child:* $\boldsymbol{\mu}_i^{\text{bg}} = \boldsymbol{\mu}_i - \frac{\lambda_{\text{2D}}}{2} s_i \mathbf{e}$, $s_i^{\text{bg}} = (1 - \lambda_{\text{2D}}) s_i$.

Both children inherit color, opacity, and spherical harmonic coefficients to maintain visual consistency.

**Recursive splitting and mask consistency.** The background child is re-evaluated in its own nearest view; if its projected ellipse still crosses another part mask, the split is applied recursively until every descendant ellipse is fully contained in a single mask. The recursion stops when the ellipse lies entirely inside its mask or the new scale falls below a minimum threshold. After each split, the children are queued in descending order of projected error, so the most inaccurate primitives are processed first. Finally, a multiview voting scheme assigns a consistent mask label to every new Gaussian, ensuring cross-view agreement.

### A.2 MESH EXTRACTION FROM THE PART-AWARE GAUSSIAN FIELD

Given the part-aware Gaussian field $\mathcal{G}^{(\boldsymbol{\alpha})} = \{\mathcal{G}_p^{(\boldsymbol{\alpha})}\}_{p=1}^{n_{\text{parts}}}$ for the interaction state $\boldsymbol{\alpha}$, we recover explicit geometry by multiview depth rendering followed by implicit–to–explicit surface reconstruction. For each subset of parts $\mathcal{G}_p^{(\boldsymbol{\alpha})}$, we render a depth map in every calibrated view, treating each Gaussian primitive as an ellipsoid whose pose and covariance are projected onto the image plane using the standard 3DGS rasterizer. The rendered depths are fused with a truncated signed-distance function (TSDF); voxel size and the truncation threshold are chosen empirically to balance detail and noise. The resulting TSDF volume provides an implicit surface for part $p$, from which we extract a triangle mesh $\mathbf{M}_p \in \mathbb{R}^{|V_p| \times 3}$ using the Marching Cubes algorithm implemented in Open3D (Huang et al., 2024). This procedure yields a high-resolution, watertight mesh for every rigid part, faithfully capturing its geometry while preserving the spatial correspondence with the underlying Gaussian field.

### A.3 JOINT TYPE IDENTIFICATION

For a part $p$ we gather Gaussian centroids in two states $P = \{p_i\}$ and $Q = \{q_i\}$, and align them with the Kabsch algorithm (Lawrence et al., 2019), which yields rotation $\mathbf{R}$ and translation $\mathbf{t}$. The residuals $d_i = q_i - (\mathbf{R}p_i + \mathbf{t})$ are stacked in the $3 \times n$ matrix $\mathbf{D} = [\, d_1 \; \cdots \; d_n\,]$, where the $n$ is the number of Gaussian kernels.

A motion of *revolute* leaves points on a common axis, so $\mathbf{D}$ is rank-1; a motion of *prismatic* rigidly translates the entire part, giving full rank. With singular values $\sigma_1 \geq \sigma_2 \geq \sigma_3$, we compute $r = (\sigma_2 + \sigma_3)/\sigma_1$. We label the joint **revolute** if $r < 0.05$ and **prismatic** otherwise, a rule that proved to be reliable across all reference objects.

### A.4 JOINT PARAMETER ESTIMATION

**Revolute joint.** A revolute joint is parameterized by a pivot point $\mathbf{p} \in \mathbb{R}^3$ and a unit quaternion $\mathbf{q} \in \mathbb{R}^4$ with $\|\mathbf{q}\| = 1$. From the Kabsch alignment (Lawrence et al., 2019) we obtain a rotation matrix $\mathbf{R} \in \mathbb{R}^{3 \times 3}$. Its rotation axis is the normalized eigenvector $\mathbf{u}$ that satisfies $\mathbf{R}\mathbf{u} = \mathbf{u}$, and the rotation angle is $\theta = \arccos\big((\mathrm{tr}(\mathbf{R}) - 1)/2\big)$. The pivot $\mathbf{p}$ is found by least-squares fitting the axis to the static centroids $\mathcal{C}_{\text{static}}$,

$$\mathbf{p} = \arg\min_{\mathbf{x}} \sum_{c \in \mathcal{C}_{\text{static}}} \|(\mathbf{x} - c) \times \mathbf{u}\|^2. \tag{15}$$

Finally, the quaternion becomes $\mathbf{q} = \big(\cos \frac{\theta}{2}, \; \mathbf{u} \sin \frac{\theta}{2}\big)$.

**Prismatic joint.** A prismatic joint is defined by a unit slide axis $\mathbf{a} \in \mathbb{R}^3$ and a translation distance $d$. Let $P = \{p_i\}$ and $Q = \{q_i\}$ be the part centroids in the two states. Their means are $\bar{p} = \frac{1}{n} \sum_i p_i$ and $\bar{q} = \frac{1}{n} \sum_i q_i$, so the rigid translation is $\mathbf{t} = \bar{q} - \bar{p}$. Hence $\mathbf{a} = \mathbf{t}/\|\mathbf{t}\|$ and $d = \|\mathbf{t}\|$. These parameters constitute the minimal motion model used in our evaluation.

## B RS-ART DATASET

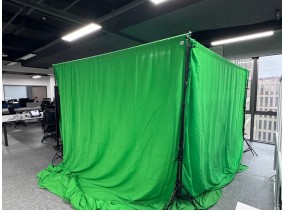 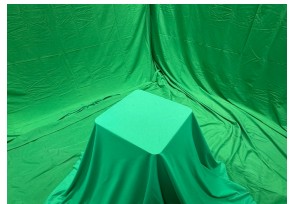 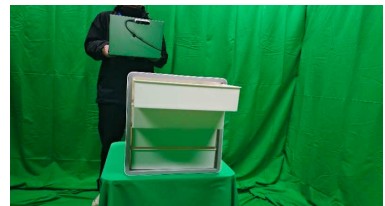

(a) Full green-screen booth     (b) Seamless pedestal platform     (c) Hand-held RGB-D capture

Figure 9: Capture platform.

**Capture platform.** Our data are acquired in a purpose-built chroma-key booth (Fig. 9). Four aluminum beams support a seamless green curtain that encloses a $2.5\,\text{m}$-high cubic volume and blocks stray reflections while maintaining uniform ambient illumination. In the center stands a square pedestal $0.5\,\text{m} \times 0.5\,\text{m}$, fully wrapped in the same green fabric. The rear side of the cloth is glued to the floor so that the pedestal merges smoothly with the ground plane, eliminating visible seams in the depth and color images.

The operator mounts an Intel RealSense RGB-D camera (Keselman et al., 2017) rigidly on the lid of a laptop, then walks a circular trajectory whose radius is chosen in real time from the live video feed to keep the target object at the desired image scale. Each circuit produces a set of synchronized RGB and depth frames that share a consistent chroma-key background, which simplifies subsequent background removal and calibration.

**Real-scene acquisition pipeline.** As illustrated in Fig. 10, to keep the real data fully compatible with the experimental settings used for the PartNet-Mobility dataset (Xiang et al., 2020), we record two complementary interaction groups for each object. (i) Multi-part sequences: all movable parts

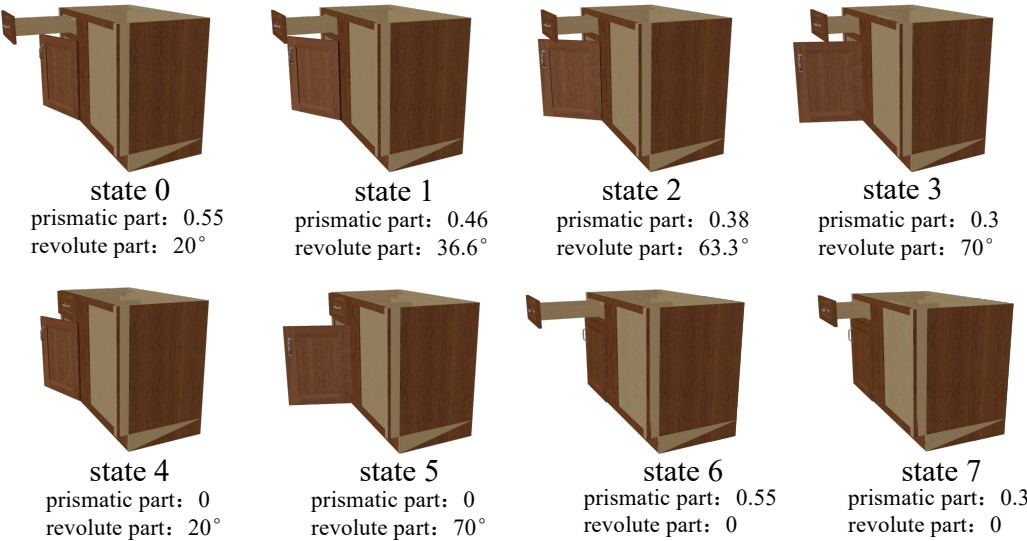

Figure 10: Example of multi-state acquisition. Using a PartNet-Mobility object for illustration, we capture eight interaction states that span the full motion range of every joint. States 0–3 (top row) depict *multi-part motion* in which all movable components change pose simultaneously, whereas States 4–7 (bottom row) show the *single-part mode* where one component moves while the others remain fixed.

are actuated simultaneously, and the capture spans four evenly spaced configurations that bracket the mechanical limits, for example, fully closed, half-open, three-quarter open, and fully open. (ii) Single-part sequences: each rotational or translational part is moved while the others remain fixed; for every part, we acquire two extreme poses corresponding to its minimum and maximum joint limits.

For each interaction state, we sample RGB-D images along a circular path whose viewing elevation is uniformly chosen in the range $30° \sim 60°$. The operator walks once around the object with an Intel RealSense camera (Keselman et al., 2017) rigidly mounted on a laptop lid, keeping the target centered in the live preview.

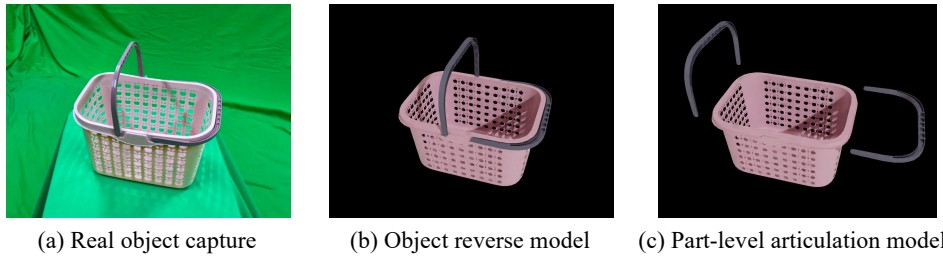

(a) Real object capture      (b) Object reverse model      (c) Part-level articulation model

Figure 11: Reverse modeling of a basket object.

**Modeling and registration workflow.** All digital assets are created in Isaac Sim 4.2 (NVIDIA, 2024a), whose USD pipeline provides both photorealistic rendering and physically consistent simulation (see Fig. 11 for a representative modeling scene). We first reverse-model each real object with calipers, producing a watertight USD mesh; subparts are manually segmented according to functional affordances and joint boundaries. High-resolution textures and PBR materials are authored in Blender 4.2.0 (Blender Foundation, 2024) and imported into Isaac Sim without loss of visual fidelity. The physical properties - mass, inertia, and friction - are assigned following the official Omniverse workflow (NVIDIA, 2024b). The joint pivots and motion limits are calibrated by system identification, ensuring that the articulation of the digital twin in Isaac Sim reproduces the joint behavior observed on the physical object. The resulting USD files provide a pixel-accurate appear-

ance and dynamics-ready articulation, forming a reliable ground truth for both vision and physics experiments.

```
RS-ART/
├── object_000/
│   ├── meta.json                  # part list, joint limits, state groups
│   ├── real/                      # multi-view RGB-D captures
│   │   ├── intrinsics.json
│   │   ├── state_00/
│   │   │   ├── rgb/000.png …
│   │   │   ├── depth/000.png …
│   │   │   └── poses.npy
│   │   ├── …
│   │   └── state_k/ …
│   ├── sim/
│   │   ├── usd/object_000.usd     # full visual+physics asset
│   │   ├── urdf/object_000.urdf   # lightweight kinematic model
│   │   └── rendered/ …            # optional re-renders, same layout as /real
│   └── meshes/                    # ground-truth part surfaces
│       ├── part_01.ply
│       └── …
├── object_001/
│   └── …                          # identical structure
└── object_017/
```

Figure 12: RS-Art Dataset directory layout.

**Dataset structure.** Fig. 12 outlines the RS-Art hierarchy. The dataset contains 18 articulated objects drawn from six everyday categories:*drawer*, *lamp*, *eyeglasses*, *floppy-disk drive*, *basket*, and *phone / laptop stand* with three instances per class. For every object we supply the complete set of multi-view RGB-D captures for each recorded interaction state together with the shared camera intrinsics and the per-view extrinsics; a fully reverse-modeling digital twin comprising a USD file that bundles part meshes, textures, physics properties, and joint definitions, a matching lightweight URDF, and watertight PLY meshes for geometry evaluation; and a concise `meta.json` that enumerates all parts, joint types, axes, limits, and labels every state as either multi-part or single-part motion. This combination of real imagery and fully articulated digital twins enables training, quantitative evaluation, and sim-to-real studies for part-aware geometry reconstruction and joint parameter estimation.

## C  EXPERIMENTAL SETTINGS

To provide the most comprehensive comparison to date, we evaluated *all* publicly available methods that match our task definition of multiview, multi-state articulated object reconstruction. Since competing pipelines differ in input assumptions and functional scope, we group them into three categories and tailor the data acquisition protocol accordingly, aiming for maximum fairness while respecting the design of each method.

**Single-part assumptions (PARIS (Liu et al., 2023) and ArticulatedGS (Guo et al., 2025)).** These methods can reconstruct only one movable part at a time. For every movable part, we capture two interaction states in which that part moves while all other parts remain fixed, and we run the pipeline independently for each part. Each state is sampled with 100 random, calibrated RGB views, giving 200 images per part.

**Multi-part, two-state methods (DTArt (Weng et al., 2024) and ArtGS(Liu et al., 2025b)).** These pipelines accept two distinct object states and can handle several movable parts jointly, provided that the part count is known. We therefore collect two interaction states and sample 100 random views per state (total 200 RGB images). The ground-truth number of movable parts is passed to both methods. As DTArt requires depth input, depth maps for every view are rendered in Gazebo (Koenig & Howard, 2004).

**Our method (PD$^2$GS).** PD$^2$GS handles an arbitrary number of parts and interaction states without knowing the part count. To match the total image budget of competing methods, we sample four

interaction states in which multiple parts move simultaneously and capture 50 random views per state, producing the same 200-image input. No depth or manual parts information is provided.

This protocol equalizes the per-run image budget to 200 views, making the inputs comparable across methods. Note, however, that single-part baselines must be executed once for the *each* movable part, so they ultimately consume $200 \times n_{parts}$ images per object.

# D    ADDITIONAL RESULTS

Table 3: Quantitative results with our method restricted to two interaction states as input.

| **Metric** | Box | Door | Eyeglasses | Faucet | Oven | Fridge | Storage | Table | Mean |
|---|---|---|---|---|---|---|---|---|---|
| Axis Ang ↓ | 1.08 | 8.70 | 0.35 | 2.40 | 1.59 | 2.64 | 2.46 | 3.70 | 2.87 |
| Axis Pos ↓ | 0.37 | 1.19 | 0.26 | 0.21 | 0.10 | 0.24 | 0.36 | 0.51 | 0.41 |
| Part Motion ↓ | 6.86 | 3.25 | 7.73 | 2.96 | 8.06 | 9.54 | 1.74 | 3.55 | 5.46 |
| CD-s ↓ | 9.21 | 6.38 | 18.55 | 1.11 | 2.16 | 14.31 | 7.64 | 4.28 | 7.96 |
| CD-m ↓ | 6.98 | 18.06 | 9.37 | 0.14 | 18.78 | 0.54 | 13.96 | 24.85 | 11.59 |
| CD-w ↓ | 8.90 | 1.73 | 11.83 | 0.54 | 9.15 | 10.43 | 4.67 | 3.75 | 6.38 |

## D.1    RESULTS ON TWO INTERACTION STATES

Because the baselines can use only two states, we limit our method to a total of 200 RGB images as input; our multi-state model therefore receives fewer views per state than the baselines, maintaining a comparable degree of fairness across methods. We add more state = 2 results for direct side-by-side comparison in Tab. 3, further strengthening the fairness of the evaluation. All reported metrics are averaged across all parts and object categories. This experiment extends Tab. 1 setting of main paper to directly assess performance under the two-state constraint. As shown in Tab. 3, our method still maintains strong performance under these settings.

Table 4: Joint-level metrics for PartNet-Mobility objects that contain three movable parts. A dash "–" marks entries that are not applicable because the corresponding joint is prismatic and therefore has no pivot position.

| Object Instance | Method | Axis Ang 0↓ | Axis Ang 1↓ | Axis Ang 2↓ | Axis Pos 0↓ | Axis Pos 1↓ | Axis Pos 2↓ | Axis Mot 0↓ | Axis Mot 1↓ | Axis Mot 2↓ |
|---|---|---|---|---|---|---|---|---|---|---|
| | PARIS (Liu et al., 2023) | 9.96 | 29.61 | 0.47 | 0.13 | 0.31 | 0.30 | 30.23 | 111.21 | 33.72 |
| | ArticulatedGS (Guo et al., 2025) | 0.52 | 6.53 | 3.62 | 0.83 | 8.32 | 0.42 | 5.23 | 2.13 | 14.43 |
| Box 102373 | DTArt (Weng et al., 2024) | 88.32 | 62.54 | 64.68 | 1.99 | 5.56 | 6.79 | 99.97 | 67.17 | 82.00 |
| | ArtGS (Liu et al., 2025b) | 0.29 | 0.05 | 0.21 | 0.19 | 0.01 | 0.39 | 15.89 | 0.17 | 6.02 |
| | PD²GS (Ours) | **0.23** | **1.34** | **0.19** | **0.03** | **0.23** | **0.14** | **1.01** | **1.59** | **2.83** |
| | PARIS (Liu et al., 2023) | 64.47 | 59.08 | 38.69 | 0.29 | 0.13 | – | 116.73 | 109.94 | 0.32 |
| | ArticulatedGS (Guo et al., 2025) | 20.43 | 4.12 | 2.32 | 1.31 | 0.21 | – | 13.11 | 3.12 | 1.31 |
| Storage 45194 | DTArt (Weng et al., 2024) | 27.60 | 0.20 | 78.01 | 4601.75 | 0.17 | – | 59.95 | 0.16 | 0.37 |
| | ArtGS (Liu et al., 2025b) | 54.17 | 1.68 | 0.75 | 0.29 | 0.08 | – | 96.54 | 1.66 | 0.02 |
| | PD²GS (Ours) | **9.82** | **1.56** | **0.56** | **0.27** | **0.03** | – | **9.79** | **1.45** | **0.07** |
| | PARIS (Liu et al., 2023) | 0.77 | 87.40 | 76.58 | – | – | 0.22 | 0.15 | 0.17 | 87.02 |
| | ArticulatedGS (Guo et al., 2025) | 1.34 | 4.24 | 2.13 | – | – | 0.73 | 0.42 | 0.14 | 0.84 |
| Storage 45271 | DTArt (Weng et al., 2024) | 4.23 | 80.96 | 74.26 | – | – | 12.52 | 0.01 | 0.19 | 50.65 |
| | ArtGS (Liu et al., 2025b) | 0.02 | 0.85 | 0.03 | – | – | 0.14 | 0.03 | 0.01 | 0.02 |
| | PD²GS (Ours) | **0.00** | **0.34** | **0.26** | – | – | **0.03** | **0.00** | **0.00** | **0.58** |
| | PARIS (Liu et al., 2023) | 1.69 | 89.40 | 79.91 | – | 0.52 | 0.84 | 0.18 | 157.14 | 109.55 |
| | ArticulatedGS (Guo et al., 2025) | 0.62 | 5.31 | 6.24 | – | 5.31 | 0.52 | 0.54 | 4.24 | 13.12 |
| Table 23372 | DTArt (Weng et al., 2024) | 0.38 | 0.22 | 0.19 | – | 0.21 | 0.01 | 0.04 | 0.14 | 0.48 |
| | ArtGS (Liu et al., 2025b) | 0.02 | 89.03 | 14.64 | – | 0.57 | 1.79 | 0.02 | 75.13 | 13.43 |
| | PD²GS (Ours) | **0.01** | **0.12** | **0.29** | – | **0.03** | **0.12** | **0.00** | **0.12** | **0.45** |

## D.2    RESULTS ON OBJECTS WITH MULTIPLE MOVABLE PARTS

Sec. 5.2 in the main paper reported our main evaluation on objects that contain two movable parts. Here, we extend the analysis to the more challenging subset, objects equipped with *three* independently actuated components, and provide both quantitative and qualitative evidence of performance.

Tab. 4 and Tab. 5 compare all baselines in the three-part instances of PartNet-Mobility (Xiang et al., 2020). Our method achieves the lowest or second–lowest error in nearly every joint metric (axis angle, axis position, axis motion) and remains highly competitive in the geometry metrics (Chamfer distances). The consistently smaller joint errors indicate that the latent–conditioned deformation field captures the complex motion couplings that arise when three parts move independently, while the geometry scores confirm that the boundary regions between parts are reconstructed sharply.

Table 5: Geometry-level metrics for PartNet-Mobility objects that contain three movable parts.

| Object Instance | Method | CD-s ↓ | CD-m 0 ↓ | CD-m 1 ↓ | CD-m 2 ↓ | CD-w ↓ |
|---|---|---|---|---|---|---|
| Box 102373 | PARIS (Liu et al., 2023) | 17.70 | 22.36 | 279.47 | 184.97 | 16.66 |
| | ArticulatedGS (Guo et al., 2025) | 16.86 | 13.59 | 23.99 | 5.18 | 13.23 |
| | DTArt (Weng et al., 2024) | 3.88 | 121.66 | 199.49 | 2.25 | 0.72 |
| | ArtGS (Liu et al., 2025b) | 9.70 | 14.18 | 5.38 | 2.63 | 7.78 |
| | PD²GS (Ours) | **6.70** | **0.88** | **0.39** | **2.21** | **5.75** |
| Storage 45194 | PARIS (Liu et al., 2023) | 28.16 | 8.77 | 193.38 | 656.10 | 176.48 |
| | ArticulatedGS (Guo et al., 2025) | 14.48 | 1.29 | 0.85 | 16.15 | 10.71 |
| | DTArt (Weng et al., 2024) | 4.41 | 71.85 | 0.09 | 33.20 | 1.12 |
| | ArtGS (Liu et al., 2025b) | 50.88 | 19.00 | 1.78 | 2041.00 | 6.77 |
| | PD²GS (Ours) | **8.88** | **0.49** | **0.51** | **4.53** | **6.04** |
| Storage 45271 | PARIS (Liu et al., 2023) | 5.09 | 668.39 | 703.55 | 19.35 | 5.50 |
| | ArticulatedGS (Guo et al., 2025) | 14.48 | 1.28 | 0.85 | 16.15 | 10.71 |
| | DTArt (Weng et al., 2024) | 2.94 | 307.65 | F | 406.23 | 8.76 |
| | ArtGS (Liu et al., 2025b) | 1.40 | 0.38 | 0.46 | 0.14 | 1.41 |
| | PD²GS (Ours) | **4.24** | **0.23** | **0.41** | **0.23** | **3.26** |
| Table 23372 | PARIS (Liu et al., 2023) | 4.14 | 5.40 | 25.67 | 45.57 | 2.03 |
| | ArticulatedGS (Guo et al., 2025) | 5.72 | 25.34 | 0.72 | 15.96 | 5.10 |
| | DTArt (Weng et al., 2024) | 1.31 | 0.37 | 0.18 | 1.21 | 0.95 |
| | ArtGS (Liu et al., 2025b) | 3.38 | 4.50 | 1.34 | 1.05 | 2.84 |
| | PD²GS (Ours) | **2.17** | **1.19** | **0.15** | **0.25** | **4.19** |

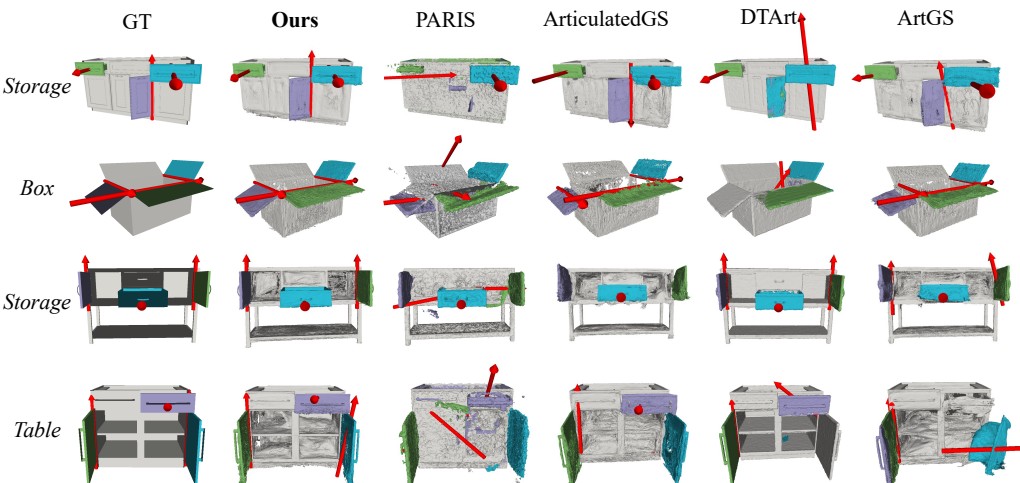

Figure 13: Qualitative multi-task modeling results on articulated objects with three movable parts.

Table 6: Evaluation of our method on object instances with five movable parts. Each reported metric is averaged across individual parts per instance.

| Object Instance | Method | Axis Ang ↓ | Axis Pos ↓ | Part Motion ↓ | CD-s ↓ | CD-m ↓ | CD-w ↓ |
|---|---|---|---|---|---|---|---|
| Storage 45271 | PARIS (Liu et al., 2023) | 61.36 | 0.41 | 48.79 | 5.86 | 380.66 | 4.96 |
| | ArtGS (Liu et al., 2025b) | 1.60 | 0.01 | 1.94 | 2.45 | 4.27 | 1.77 |
| | PD²GS (Ours) | **0.02** | **0.00** | **0.02** | **1.39** | **0.71** | **1.25** |
| Storage 45612 | PARIS (Liu et al., 2023) | 61.37 | 0.14 | 46.16 | 5.89 | 577.17 | 5.07 |
| | ArtGS (Liu et al., 2025b) | 0.11 | 0.08 | 1.82 | 2.53 | 4.09 | 1.83 |
| | PD²GS (Ours) | **0.04** | **0.03** | **0.01** | **1.40** | **0.53** | **1.32** |
| Table 30666 | PARIS (Liu et al., 2023) | 36.58 | - | 0.30 | 7.73 | 361.76 | 6.52 |
| | ArtGS (Liu et al., 2025b) | 8.91 | - | 0.15 | 9.68 | 1159.34 | 5.70 |
| | PD²GS (Ours) | **0.15** | **-** | **0.05** | **4.05** | **1.47** | **3.65** |
| Table 33810 | PARIS (Liu et al., 2023) | 1.07 | 0.03 | **0.92** | **3.53** | 73.97 | 3.44 |
| | ArtGS (Liu et al., 2025b) | 48.67 | 32.02 | 8.04 | 5.75 | 1346.58 | **3.18** |
| | PD²GS (Ours) | **0.44** | **0.02** | 0.93 | 4.85 | **0.68** | 3.91 |

Fig. 13 compares our method with the baselines on the objects with three movable parts and visualizes *all* recovered outputs: part-aware geometry (colored meshes), joint axes (red arrows) and part labels. The results demonstrate that our deformable-field formulation yields a coherent solution across all three modalities: the reconstructed geometry is complete and well aligned, the estimated

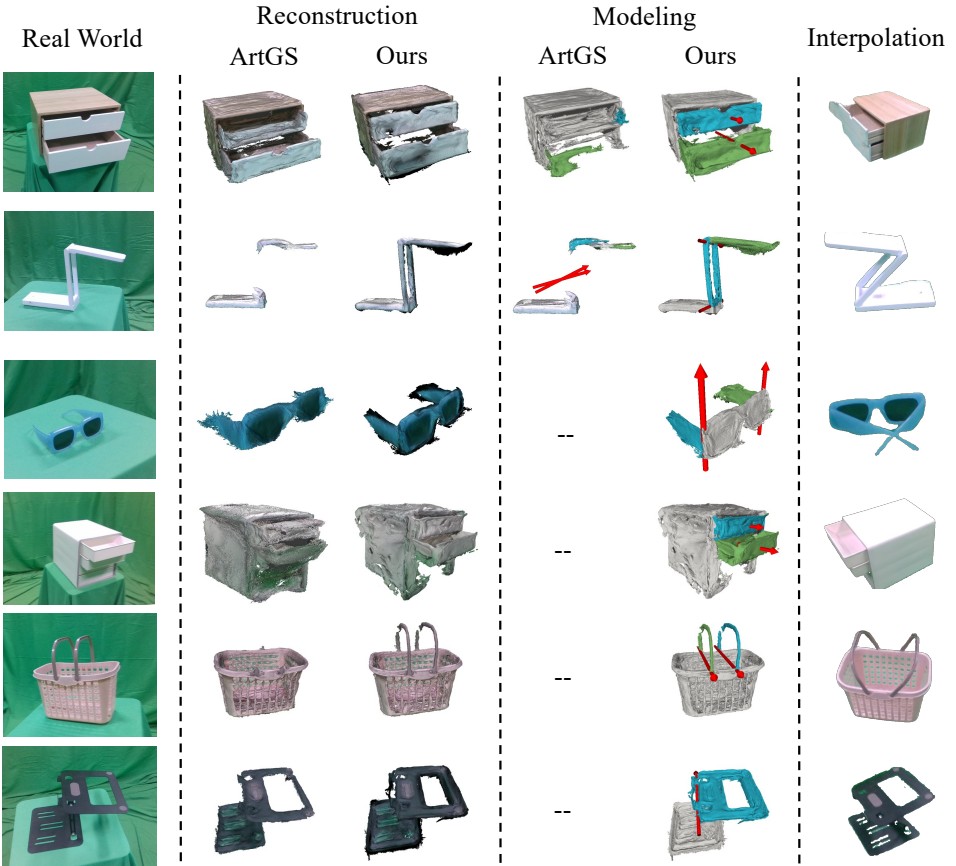

Figure 14: Qualitative comparison on real objects from RS-Art.

axes coincide with the true joint locations and orientations, and the segmentation cleanly separates the three movable parts. Together, these visual cues indicate that the latent-conditioned deformation field is able to discover and model complex kinematic structures, even with three interacting joints, without manual specification of part identities.

To further validate scalability to higher articulation complexity, Tab. 6 reports additional results on objects with five movable parts. These instances present substantially more challenging kinematic couplings and occlusion patterns than the two- and three-part cases. Despite the increased difficulty, our approach continues to produce accurate joint estimates and detailed geometry.

### D.3    RESULTS ON THE RS-ART DATASET

Fig. 14 and Tab. 7 present qualitative and quantitative results on our RS-Art dataset. Although the capture setup avoids strong specular highlights and cast shadows, the real objects still exhibit complex materials, weak surface texture, and a narrower viewing envelope than in simulation, making the task substantially harder than on PartNet-Mobility (Xiang et al., 2020). Among the baselines, only ArtGS can be made to run reliably on the real data, so we compare against it on every instance that yields a converged model. Across all evaluated objects, our method achieves lower joint parameter and geometry errors and produces cleaner part segmentation, confirming that the deformation-field formulation transfers to real scenes. Meanwhile, the results highlight the open challenges discussed in Sec. E: fine surface details degrade under limited texture and grazing angles, and heavy occlusion can still disturb the 3DGS reconstruction. These observations underline the necessity of RS-Art as a bridge between synthetic evaluation and real-world deployment.

Fig. 15 shows qualitative results on RS-Art objects captured in real scenes, where cluttered backgrounds, natural illumination, and real-world materials make reconstruction noticeably more chal-

Table 7: Results on the RS-Art Dataset.

| Metric | Method | Real-World Objects | | | | | | Mean |
| | | Drawers | Desk Lamps | Eyeglasses | Floppy-Disk Drives | Woven Baskets | Phone/Laptop Stand | |
|---|---|---|---|---|---|---|---|---|
| Axis ↓ Ang 0 | ArtGS (Liu et al., 2025b) | 41.86 | 72.98 | 54.24 | 29.24 | 45.56 | 63.54 | 51.24 |
| | PD²GS (Ours) | **0.84** | **0.32** | **4.45** | **1.71** | **0.88** | **0.54** | **1.46** |
| Axis ↓ Ang 1 | ArtGS (Liu et al., 2025b) | 3.43 | 83.25 | 23.24 | 83.24 | 57.24 | 24.64 | 45.84 |
| | PD²GS (Ours) | **0.25** | **0.34** | **0.90** | **1.08** | **0.58** | **15.27** | **3.07** |
| Axis ↓ Pos 0 | ArtGS (Liu et al., 2025b) | – | 8.89 | 2.54 | – | 352.35 | 35.77 | 99.89 |
| | PD²GS (Ours) | – | **0.36** | **2.07** | – | **0.89** | **1.17** | **1.12** |
| Axis ↓ Pos 1 | ArtGS (Liu et al., 2025b) | – | 8.56 | 12.23 | – | 83.35 | 45.35 | 37.37 |
| | PD²GS (Ours) | – | **0.32** | **0.36** | – | **0.40** | **9.31** | **2.60** |
| Part ↓ Motion 0 | ArtGS (Liu et al., 2025b) | 36.75 | 14.94 | 9.12 | 14.56 | 92.69 | 14.35 | 30.40 |
| | PD²GS (Ours) | **0.58** | **0.87** | **0.63** | **1.62** | **9.46** | **1.79** | **2.49** |
| Part ↓ Motion 1 | ArtGS (Liu et al., 2025b) | 16.38 | 13.07 | 43.33 | 63.45 | 150.35 | 5.35 | 48.65 |
| | PD²GS (Ours) | **0.36** | **0.58** | **0.74** | **2.27** | **4.97** | **5.04** | **2.33** |
| CD-s ↓ | ArtGS (Liu et al., 2025b) | 41.39 | 11.38 | 25.24 | 53.53 | 6.34 | 14.35 | 25.37 |
| | PD²GS (Ours) | **13.13** | **6.25** | **10.19** | **26.88** | **4.05** | **13.36** | **12.31** |
| CD-m 0 ↓ | ArtGS (Liu et al., 2025b) | 121.69 | 39.09 | 33.24 | 104.35 | 58.24 | 26.49 | 63.85 |
| | PD²GS (Ours) | **3.09** | **4.83** | **7.19** | **10.37** | **5.63** | **10.08** | **6.86** |
| CD-m 1 ↓ | ArtGS (Liu et al., 2025b) | 180.57 | 34.18 | 234.24 | 156.35 | 62.64 | 43.98 | 118.66 |
| | PD²GS (Ours) | **8.57** | **3.86** | **9.41** | **13.47** | **0.27** | **41.44** | **12.84** |
| CD-w ↓ | ArtGS (Liu et al., 2025b) | **7.96** | 6.09 | 20.35 | 96.35 | 36.25 | 35.96 | 33.83 |
| | PD²GS (Ours) | 16.55 | **3.05** | **14.80** | **19.03** | **12.78** | **29.90** | **14.35** |

Figure 15: Qualitative comparison on real-world scenes.

lenging. Even under these conditions, our method recovers sharper geometry and more stable part motions, particularly when object boundaries remain reasonably clear. To ensure a fair evaluation, we also conduct reconstruction comparisons on the real-scene benchmark of the PARIS dataset. In these challenging scenarios where baselines often suffer from severe structural degradation or fail to converge, PD²GS consistently maintains structural integrity and cleaner part boundaries, producing practically usable reconstructions while suppressing artifacts such as surface blurring and deformation spillover.

Table 8: Robustness evaluation under camera pose inaccuracies.

| Method | Axis Ang ↓ | Axis Pos ↓ | Part Motion ↓ | CD-s ↓ | CD-m ↓ | CD-w ↓ |
|---|---|---|---|---|---|---|
| PARIS (Liu et al., 2023) w/o Perturbation | 29.98 | 0.38 | 70.84 | 12.98 | 103.22 | 10.95 |
| ArtGS (Liu et al., 2025b) w/o Perturbation | 3.53 | 3.46 | 5.49 | 4.29 | 82.65 | 6.72 |
| PD²GS (Ours) w/ (1.0°, 0.02) | 2.92 | 0.22 | 3.66 | 11.02 | 11.09 | 7.05 |
| PD²GS (Ours) w/ (3.0°, 0.05) | 10.00 | 0.24 | 1.55 | 13.60 | 24.71 | 9.19 |
| PD²GS (Ours) w/ colmap | 1.93 | 0.11 | 3.85 | 7.39 | 6.82 | 7.63 |
| PD²GS (Ours) w/o Perturbation | **0.40** | **0.06** | **0.43** | **2.05** | **1.12** | **3.02** |

## D.4 EFFECT OF CAMERA POSE ERRORS

To evaluate robustness to camera pose inaccuracies, we design three controlled perturbation settings. Starting from ground-truth poses, we inject Gaussian noise into the rotation and translation components to form two perturbation levels, denoted as $(u, v)$, where $u$ is the standard deviation

of rotational noise measured in degrees and $v$ is that of translational noise measured in scene units. In addition, we test a third setup that uses camera poses estimated by COLMAP. All methods are evaluated using the same protocol as Tab. 1 in the main paper. Metrics are averaged across all articulated parts and objects. As shown in Tab. 8, although performance decreases slightly due to these perturbations, our method consistently maintains superior results compared to prior methods.

Table 9: Quantitative comparison with articulated generation baselines on the Storage category.

| Object | Method | Axis Ang ↓ | Axis Pos ↓ | Part Motion ↓ | CD-s ↓ | CD-m ↓ | CD-w ↓ |
|---|---|---|---|---|---|---|---|
| | NAP (Lei et al., 2023) | 3.88 | 2.45 | 7.29 | 5.56 | 5.19 | 8.54 |
| Storage | SINGAPO (Liu et al., 2025a) | 2.72 | 0.86 | 9.03 | - | - | - |
| | PD$^2$GS (Ours) | **0.13** | **0.02** | **0.32** | **2.78** | **0.07** | **5.48** |

## D.5 Comparison with articulated generation methods

We further extended our evaluation to include generative articulated object models, NAP (Lei et al., 2023) and SINGAPO (Liu et al., 2025a). NAP (Lei et al., 2023) operates by generating target objects from Gaussian noise derived from specific instances, whereas SINGAPO (Liu et al., 2025a) infers them from a single input view. To ensure a robust evaluation for SINGAPO (Liu et al., 2025a) given its single-view dependence, we randomly sampled 50 input views for each object and reported the metrics from the best-performing result. As shown in Tab. 9, both methods exhibit significant deviations in joint motion analysis compared to our approach, reflecting the inherent distinction between generative plausibility and the precise kinematic inference required for our task.

Table 10: Training and inference time for objects with varying numbers of articulated parts.

| Number of parts | Method | 2 | 3 | 5 |
|---|---|---|---|---|
| | PARIS (Liu et al., 2023) | 31.38 | 45.99 | 75.15 |
| Training (min) | DTArt (Weng et al., 2024) | 26.31 | 31.42 | 43.76 |
| | ArtGS (Liu et al., 2025b) | 16.22 | 15.08 | 17.86 |
| | PD$^2$GS (Ours) | 35.59 | 38.20 | 44.82 |
| | PARIS (Liu et al., 2023) | 24773.60 | 37180.59 | 61924.30 |
| Inference (ms) | DTArt (Weng et al., 2024) | 568.67 | 525.70 | 613.30 |
| | ArtGS (Liu et al., 2025b) | 3.42 | 3.71 | 4.16 |
| | PD$^2$GS (Ours) | 37.93 | 40.86 | 37.22 |

## D.6 Results on Training and Inference Time

We provide statistics on training and inference overhead for reference, as shown in Tab. 10. For a fair comparison, all methods are trained on the same dataset of 200 views per object. Regarding inference, since discrete-state baselines (DTArt (Weng et al., 2024), ArtGS (Liu et al., 2025b)) cannot natively infer arbitrary continuous states, we measured their timings by explicitly rigidly transforming their reconstructed models using inferred joint parameters. Specifically, for ArtGS (Liu et al., 2025b), we render by rigidly transforming the reconstructed Gaussian field; thus, the inference time is equivalent to standard static 3DGS (Kerbl et al., 2023). In contrast, our method requires the evaluation of a latent-conditioned deformation field. In the context of articulated object modeling, inference efficiency is not the primary limiting factor. Thus, this difference is secondary to the substantial gains in modeling capability.

## D.7 Ablation Studies

**Refinement stage.** Sec. 5.5 in the main paper visualizes the qualitative benefit of coarse-to-fine refinement. Tab. 11 complements that visual evidence with numerical results on four representative objects. Across all geometry metrics, the refined model (*w/ Refined*) consistently outperforms the same network trained without the refinement stage (*w/o Refined*). Averaged across the four objects, the refinement reduces CD-s by $35\%$ and CD-m by $60\%$. The greatest gains occur at part boundaries (CD-m columns), confirming that boundary-aware Gaussian splitting and local fine-tuning

Table 11: Quantitative ablation study on the refinement stage.

| Object Instance | Method | CD-s ↓ | CD-m 0 ↓ | CD-m 1 ↓ | CD-w ↓ |
|---|---|---|---|---|---|
| Box 100676 | w/o Refined | 11.66 | 2.12 | 2.14 | 10.16 |
| | w/ Refined | **9.42** | **1.58** | **1.52** | **8.03** |
| Oven 7187 | w/o Refined | 21.56 | 5.44 | 2.16 | 17.58 |
| | w/ Refined | **16.58** | **0.32** | **0.54** | **13.91** |
| Refrigerator 12248 | w/o Refined | 16.94 | 0.60 | 0.92 | 12.06 |
| | w/ Refined | **13.63** | **0.21** | **0.23** | **9.53** |
| Storage 47254 | w/o Refined | 13.63 | 2.91 | 1.24 | 9.53 |
| | w/ Refined | **6.52** | **0.60** | **0.92** | **5.05** |

Table 12: Ablation on the number of interaction states $K$. Each column lists the performance metrics obtained with the corresponding $K$, evaluated on the PartNet-Mobility object Door 9168.

| **Metric** | $K=2$ | $K=4$ | $K=6$ | $K=8$ | $K=10$ |
|---|---|---|---|---|---|
| Axis Ang 0 ↓ | 31.45 | 0.30 | 0.14 | **0.07** | 0.36 |
| Axis Ang 1 ↓ | 47.83 | 0.33 | 0.22 | 0.17 | **0.15** |
| Axis Pos 0 ↓ | 0.38 | 0.34 | 0.12 | **0.06** | 0.18 |
| Axis Pos 1 ↓ | 0.07 | 0.26 | 0.19 | **0.05** | 0.29 |
| Part Motion 0 ↓ | 57.60 | 1.44 | 1.35 | 1.24 | **1.03** |
| Part Motion 1 ↓ | 56.71 | 2.38 | 1.74 | **0.71** | 2.10 |
| CD-s ↓ | 1.63 | **0.53** | 0.56 | 0.63 | 0.77 |
| CD-m 0 ↓ | 85.87 | 0.63 | **0.49** | 1.38 | 1.24 |
| CD-m 1 ↓ | 142.74 | **0.33** | 0.57 | 0.90 | 0.79 |
| CD-w ↓ | 0.81 | **0.60** | 0.64 | 0.84 | 0.87 |

markedly sharpen the reconstructed surfaces and prevent inter-penetration. These numbers corroborate the qualitative observations in Sec. 5.5 and highlight the refinement stage as a key component for accurate part-level geometry.

$K = 2$  $K = 4$  $K = 6$  $K = 8$  $K = 10$

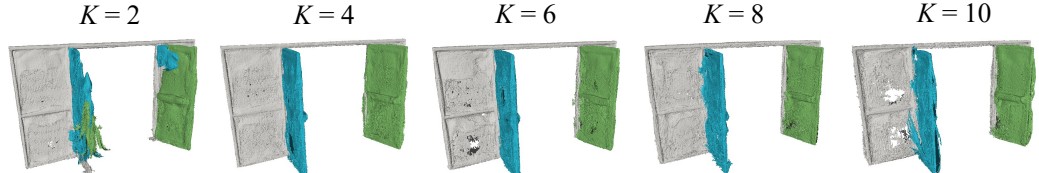

Figure 16: Qualitative effect of the number of interaction states.

**The number of interaction states.** Tab. 12 and Fig. 16 show how our method reacts to different numbers of input interaction states $K$. With just two interaction states ($K = 2$), the network sees only a start-to-end displacement for each Gaussian. This limited evidence allows multiple trajectory assignments to fit the same images, so Gaussians can drift between neighboring parts, leading to inflated joint errors and indistinct part boundaries. As more states are provided, every primitive must follow a longer and more distinctive trajectory. The stronger constraint eliminates ambiguity: a Gaussian that does not truly belong to a part can no longer match the motion of that part in all states, and the optimizer moves it to the correct rigid component, causing a rapid drop in error. However, after each joint has appeared in several diverse poses, additional states add little fresh kinematic information while still introducing measurement noise and extra training time, so the accuracy curve levels off and may fluctuate slightly.

**Prompting strategies on component-count estimation.** Tab. 13 compares our approach with two alternative prompting strategies, specifically single-view queries and generic cross-state queries, and shows that our method provides the most stable and accurate component-count estimates. VLM-M1 queries single views ("How many movable components does this object contain?"); VLM-M2

Table 13: Movable-component count estimation accuracy (correct/total number) for three VLM prompting schemes across eight object categories (instance counts in parentheses).

| Method | Box (2) | Door (3) | Eyeglasses (2) | Faucet (2) | Oven (2) | Fridge (3) | Storage (3) | Table (3) | Overall Accuracy |
|---|---|---|---|---|---|---|---|---|---|
| VLM-M1 | 1 / 2 | 3 / 3 | 2 / 2 | 1 / 2 | 2 / 2 | 3 / 3 | 3 / 3 | 3 / 3 | 90 % |
| VLM-M2 | 0 / 2 | 0 / 3 | 1 / 2 | 2 / 2 | 0 / 2 | 0 / 3 | 0 / 3 | 0 / 3 | 15 % |
| VLM-Ours | 2 / 2 | 3 / 3 | 2 / 2 | 2 / 2 | 2 / 2 | 3 / 3 | 3 / 3 | 3 / 3 | **100 %** |

Table 14: Comparison of SAM-based segmentation accuracy between our approach and the SAGD method.

| Metric | Method | Box | Door | Eyeglasses | Faucet | Oven | Fridge | Storage | Table | Mean |
|---|---|---|---|---|---|---|---|---|---|---|
| IoU | SAGD (Hu et al., 2024) | 45.37 | 42.79 | 28.49 | 34.71 | 38.32 | 45.02 | 44.70 | 58.04 | 42.18 |
| | SAM-Ours | **84.20** | **89.61** | **69.40** | **83.72** | **75.33** | **87.09** | **78.06** | **79.25** | **80.83** |
| Acc | SAGD (Hu et al., 2024) | 94.63 | 97.14 | 97.96 | 97.46 | 91.64 | 95.23 | 94.54 | 97.79 | 95.80 |
| | SAM-Ours | **98.06** | **98.75** | **99.57** | **99.47** | **98.70** | **99.37** | **97.92** | **98.76** | **98.82** |
| F1-score | SAGD (Hu et al., 2024) | 54.46 | 52.34 | 38.01 | 49.02 | 46.89 | 52.11 | 55.31 | 65.21 | 51.67 |
| | SAM-Ours | **89.74** | **95.05** | **81.04** | **90.82** | **81.42** | **91.46** | **85.00** | **85.58** | **87.51** |

queries cross-state pairs ("How many movable parts are identifiable from these two states of the same object?"). Final counts are determined by the most frequent response. This design eliminates the need for manual $n_{parts}$ selection required in previous 3DGS pipelines.

**SAM-based component segmentation.** For SAM segmentation, as shown in Tab. 14, we compare our approach to the current state-of-the-art point-prompt SAM-based 3DGS segmentation method, SAGD (Hu et al., 2024). In all cases, these alternatives perform worse than the full pipeline, providing quantitative evidence for the value of each component.

Table 15: Quantitative evaluation of each pipeline module on all object categories. All metrics are averaged over parts. For Acc and IoU (reported as %), higher values are better; for CD-w (reported in mm), lower values are better.

| Module & Metric | Box | Door | Eyeglasses | Faucet | Oven | Fridge | Storage | Table | Mean |
|---|---|---|---|---|---|---|---|---|---|
| Deformable GS (CD-w) | 8.32 | 0.64 | 0.27 | 0.57 | 13.78 | 9.43 | 3.94 | 3.05 | 5.00 |
| Coarse Seg. (Acc, %) | 72.63 | 86.93 | 86.45 | 66.82 | 59.45 | 67.57 | 67.64 | 82.45 | 73.74 |
| SAM+Prompt (IoU, %) | 84.20 | 89.61 | 69.40 | 83.72 | 75.33 | 87.09 | 78.06 | 79.25 | 80.83 |
| Refined Seg. (Acc,%) | 95.83 | 95.41 | 98.64 | 99.53 | 99.23 | 96.41 | 85.76 | 98.29 | 96.14 |

**Each pipeline module.** For each key module, we report a dedicated quantitative metric in the Tab. 15. For Deformable GS reconstruction(Sec. 3.2), Chamfer distance demonstrates stable and accurate geometric modeling across all categories. For VLM-based part counting (Sec. 3.3), Delivers consistently high accuracy, validating the effectiveness of our task-specific prompt. For Coarse Gaussian Segmentation (Sec. 3.3), High accuracy confirms reliable part-level grouping, enabling robust downstream point prompt for SAM. For SAM with prompt (Sec. 3.4), Achieves precise semantic segmentation. For Refined segmentation(Sec. 3.4), Post-refinement clustering further increases accuracy, showing precise decoupling of part-level Gaussians. The consistency of these metrics across all objects and scenes confirms that each module is robust and that the pipeline as a whole is stable.

# E LIMITATIONS

**Incomplete part reconstruction under occlusion.** Like other NeRF-based and 3D Gaussian Splatting-based approaches, PD$^2$GS may struggle to reconstruct complete geometry and texture for every part of the object. This limitation arises in cases of severe occlusion, limited viewpoints, or coupled articulation, where certain surfaces are never observed. In addition, the representation is not surface-aware by design, with both the NeRF and 3DGS models radiance and density fields rather than explicit meshes, which can lead to surface artifacts and coarse or noisy geometry. Real-scene results also suggest that texture fidelity can degrade under challenging lighting conditions. Future

work may explore integrating surface priors, reflectance models, or learned data-driven priors, such as diffusion models, to enhance reconstruction fidelity and plausibly complete unobserved geometry in under-constrained regions.

**Limited articulation understanding from sparse states.** PD$^2$GS infers motion by comparing interaction states, but reasoning about joint types and motion limits remains difficult. In particular, screw-type joints are difficult to identify, as the motion between observed states may produce little or no visible appearance change, making them indistinguishable from static or simpler joints under visual comparison alone. Similarly, estimating articulation limits (e.g., joint range or end stops) requires observing motions near those extremes, which sparse or biased sampling may not capture. Incorporating additional interaction sequences or learning from temporal transitions could help enrich articulation inference in future extensions.

**Lack of physical properties for simulation.** While PD$^2$GS focuses on visual and kinematic modeling, it does not capture physical properties such as mass, friction, or compliance, which are essential for high-fidelity digital twins. These properties are crucial for accurate simulation and downstream robotic tasks, but are currently absent from most NeRF/3DGS-based pipelines. Bridging this gap may require integrating material prediction models, learning from physical interactions, or coupling neural reconstruction with differentiable simulation frameworks.

