# OpenReview forum: "PD$^{2}$GS: Part-Level Decoupling and Continuous Deformation of Articulated Objects via Gaussian Splatting"
_ICLR.cc/2026/Conference — ICLR 2026 Poster_

### Official Review · Reviewer_HeoQ · 2025-10-19

**Soundness:** 3
**Presentation:** 3
**Contribution:** 2
**Rating:** 6
**Confidence:** 3

**Summary:**

This paper introduces PD²GS, a novel framework for reconstructing and modeling articulated objects from multi-view images without manual supervision. Its core idea is to represent an object's various interaction states as continuous deformations of a single, shared canonical 3D Gaussian field, enabling smooth control and interpolation. Key contributions include a coarse-to-fine segmentation that automatically discovers rigid parts by clustering motion trajectories and refining boundaries with SAM, and the release of the RS-Art dataset for real-world evaluation.

**Strengths:**

The core idea of modeling all interaction states as continuous deformations of a single, shared canonical 3D Gaussian field is both simple and powerful. This elegantly sidesteps the "representational fragmentation" of prior two-state methods, enabling smooth, continuous control and interpolation of articulated poses, which is a major step towards high-fidelity digital twins. The framework automatically infers the number and boundaries of rigid parts without manual supervision. It achieves this through a clever coarse-to-fine process that first clusters Gaussians by their motion trajectories (using a VLM for part counting) and then refines boundaries using SAM, making it highly applicable to real-world objects with unknown kinematics.

**Weaknesses:**

The empirical evaluation lacks comparison to foundational dynamic scene representation works like D²NeRF (Dynamically Deformable NeRF) or Gao et al.'s deformable 3DGS, which also model scenes via a canonical field and latent-code-driven deformation. This omission makes it difficult to assess the true novelty and contribution of the deformation modeling component beyond the specific task of articulation.
The method is explicitly noted to assume "accurate camera poses," and its robustness to pose estimation noise, a common issue in real-world applications, remains entirely unvalidated. This is a significant practical limitation that is not addressed through ablations or sensitivity analysis, casting doubt on the method's real-world readiness. While tested on objects with up to three parts, there is no evidence provided for the method's performance on objects with a higher number of articulated parts (e.g., >5). The clustering and segmentation pipeline may face challenges with increasing complexity, and its scalability remains an open and significant question.

**Questions:**

1.Dynamic Scene Baselines: Why were foundational dynamic scene representations like D²NeRF or other deformable 3DGS methods not included as baselines? A comparison would help clarify whether the performance gains are specific to the articulated object modeling pipeline or also represent a general advance in deformation field modeling.

2.Camera Pose Robustness: The paper states an assumption of accurate camera poses. Could you provide an ablation or sensitivity analysis on the robustness of PD²GS to noisy camera poses, which are common in real-world SfM pipelines? This would significantly strengthen the claim of real-world applicability.

3.Handling Occlusion: The limitation of being unable to reconstruct unobserved geometry is acknowledged. Have the authors considered or experimented with incorporating learned or data-driven priors (e.g., diffusion models, symmetry) to plausibly complete the occluded parts of an object, especially around joints?

---

> ### Author Response · Authors · 2025-11-22
> **Response to Reviewer HeoQ 1/2**
>
> We sincerely appreciate the reviewer's assessment of our work, particularly noting the simplicity and effectiveness of modeling articulated states as continuous deformations of a shared canonical Gaussian field, and highlighting our coarse-to-fine segmentation approach for automatically discovering rigid parts without supervision.
>
> In response to the specific comments and suggestions raised by the reviewer, we clarify as follows:
>
> > **W1&Q1:** Dynamic Scene Baselines.
>
> **A1:** We thank the reviewer's suggestion regarding comparisons to dynamic scene models such as D$^2$NeRF[1] and deformable 3D Gaussians[2]. Please excuse us for not finding papers exactly matching the titles provided by the reviewer; instead, we identified two closely related works with similar titles and topics. If these differ from those intended by the reviewer, please kindly let us know.
>
> Our key contribution is centered on articulated multi-part object modeling rather than proposing a universally superior deformation framework. Our deformation module is specifically designed to achieve a unified and continuous Gaussian field representation that enables explicit, controllable, and decoupled articulation of multi-part objects, capabilities that general dynamic scene methods cannot achieve. Thus, general dynamic-scene methods were not included as direct baselines, as they primarily focus on continuous deformation and novel-view rendering without explicitly addressing articulated multi-part object modeling and motion analysis. For instance, explicitly specifying continuous motions of different parts, as demonstrated in our [video](https://pd2gs-iclr.github.io/pd2gs-iclr/ ), is beyond the scope of these methods.
> + The baseline we included (ArticulatedGS)[3] in the Tab.1 of main paper is built upon deformable 3D Gaussians, and our results consistently outperform it in our specific task settings.
> + We compared our method with D$^2$NeRF[1] on rendering novel intermediate states, tested on four "Storage" object instances with four interaction states each. We evaluated interpolated states between consecutive input states, as summarized in Table 1, demonstrating our clear advantage on all metrics.
>
> We will explicitly discuss these distinctions and clarify the different representation goals in the revision.
>
> *Table 1: Novel-state rendering comparison with D$^2$NeRF on the "Storage" objects.*
> | **Method** | **PSNR ↑** | **SSIM ↑** | **LPIPS ↓** |
> |:---:|:---:|:---:|:---:|
> | D$^2$NeRF | 21.34 | 0.85 | 0.16 |
> | Ours | **28.02** | **0.93** | **0.09** |
>
> [1] Wu, Tianhao, et al. "D$^2$nerf: Self-supervised decoupling of dynamic and static objects from a monocular video." Advances in neural information processing systems 35 (2022): 32653-32666.
>
> [2] Yang, Ziyi, et al. "Deformable 3d gaussians for high-fidelity monocular dynamic scene reconstruction." Proceedings of the IEEE/CVF conference on computer vision and pattern recognition. 2024.
>
> [3] Guo, Junfu, et al. "Articulatedgs: Self-supervised digital twin modeling of articulated objects using 3d gaussian splatting." Proceedings of the Computer Vision and Pattern Recognition Conference. 2025.
>
> > **W2&Q2:** Accurate camera poses.
>
> **A2:** We appreciate the reviewer’s valuable suggestion regarding robustness to camera pose inaccuracies. Our RS-Art experiments on real-world objects were conducted without ground-truth camera poses, yet we obtained consistently stable and accurate results, as shown in Tab.5 of the appendix.
>
> To further validate robustness, we performed three additional experiments by introducing camera pose perturbations with different noise levels and colmap without GTs. The results (Table 2 below) confirm that our approach remains stable under realistic pose errors.
>
> *Table 2:Robustness evaluation under camera pose perturbations. (u, v) indicates standard deviations of rotation perturbation and translation perturbation respectively. Metrics are averaged across parts and objects, consistent with Table 1 of the main paper.*
> | **Method** | **Axis Ang ↓** | **Axis Pos ↓** | **Part Motion ↓** | **CD-s ↓** | **CD-m ↓** | **CD-w ↓** |
> |:---:|:---:|:---:|:---:|:---:|:---:|:---:|
> | PARIS w/o Perturbation | 29.98 | 0.38 | 70.84 | 12.98 | 103.22 | 10.95 |
> | ArtGS w/o Perturbation | 3.53 |3.46|5.49| 4.29 | 82.65 | 6.72 |
> | Ours w/ (1.0°, 0.02 ) | 2.92  | 0.22 | 3.66 | 11.02 | 11.09 | 7.05 |
> | Ours w/ (3.0°, 0.05)  | 10.00 | 0.24 | 1.55 | 13.60 | 24.71 | 9.19 |
> | Ours w/ colmap        | 1.93  | 0.11 | 3.85 | 7.39  | 6.82  | 7.63 |
> | Ours w/o Perturbation | **0.40**  | **0.06** | **0.43** | **2.05**  | **1.12**  | 3.02 |

---

> ### Author Response · Authors · 2025-11-22
> **Response to Reviewer HeoQ 2/2**
>
> > **W3** Performance on objects with a higher number of articulated parts.
>
> **A3:** Our paper already evaluates performance across a wider variety of categories and real-world scenarios than prior work, demonstrating the robustness of our method. We add additional results below for objects with five movable parts, showing that our method continues to achieve strong performance even in these more complex scenarios.
>
> *Table 3: Evaluation of our method on randomly selected object instances with five articulated parts. Each reported metric is averaged across individual parts per instance.*
> | **Object Instance** | **Method** | **Axis Ang ↓** | **Axis Pos ↓** | **Part Motion ↓** | **CD-s ↓** | **CD-m ↓** | **CD-w ↓** |
> |:---:|:---:|:---:|:---:|:---:|:---:|:---:|:---:|
> | Storage 45271 | PARIS | 61.36 | 0.41 | 48.79 | 5.86 | 380.66 | 4.96 |
> | | ArtGS | 1.60 | 0.01 | 1.94 | 2.45 | 4.27 | 1.77 |
> | | Ours | **0.02** | **0.00** | **0.02** | **1.39** | **0.71** | **1.25** |
> | Storage 45612 | PARIS | 61.37 | 0.14 | 46.16 | 5.89 | 577.17 | 5.07 |
> | | ArtGS | 0.11 | 0.08 | 1.82 | 2.53 | 4.09 | 1.83 |
> | | Ours | **0.04** | **0.03** | **0.01** | **1.40** | **0.53** | **1.32** |
> | Table 30666 | PARIS | 36.58 | - | 0.30 | 7.73 | 361.76 | 6.52 |
> | | ArtGS | 8.91 | - | 0.15 | 9.68 | 1159.34 | 5.70 |
> | | Ours | **0.15** | - | **0.05** | **4.05** | **1.47** | **3.65** |
> | Table 33810 | PARIS | 1.07 | 0.03 | **0.92** | **3.53** | 73.97 | 3.44 |
> | | ArtGS | 48.67 | 32.02 | 8.04 | 5.75 | 1346.58 | **3.18** |
> | | Ours | **0.44** | **0.02** | 0.93 | 4.85 | **0.68** | 3.91 |
>
> > **Q3** Handling Occlusion.
>
> **A4:** We fully agree with the reviewer's insightful comment regarding handling occluded geometry. Our paper's Limitations section already notes that PD$^2$GS, like other NeRF/3DGS methods, may fail to recover complete geometry for parts that are never fully seen . So far, we have taken a conservative approach: we only model what is observed, and did not attempt to hallucinate occluded surfaces. The rationale was to avoid introducing guesswork that might reduce accuracy of the visible regions. Meanwhile, we agree that incorporating priors to infer likely shapes for hidden areas is a promising direction for increasing completeness.

---

### Official Review · Reviewer_5MT6 · 2025-10-27

**Soundness:** 3
**Presentation:** 3
**Contribution:** 3
**Rating:** 6
**Confidence:** 4

**Summary:**

The paper addresses the problem of reconstructing articulated objects from multi-view, multi-state observations. The approach first learns a smooth deformation of a shared canonical field for each interaction state, and then uses the resulting deformation trajectories to guide a progressive coarse-to-fine part segmentation. The segmentation is further refined using SAM-based cues and boundary-aware Gaussian splitting. The method then estimates per-part meshes as well as joint types and parameters. In addition, the paper introduces a new dataset, RS-Art, containing a large number of real-world captures of articulated objects.

**Strengths:**

1. The newly proposed dataset RS-Art should be useful for further research work if made public, especially those real-world captures.
2. The paper seems to achieve SOTA performance than baselines with multi-state multi-view images in most cases.
3. The authors conducted extensive experiments on different datasets.

**Weaknesses:**

1. The whole systems seem to compose of numerous parts, which may be a little complicate and hard to extend.
2. Some visualizations on the newly-proposed dataset, including the data itself and the reconstructed results in videos would help readers grasp the new dataset.
3. The proposed method seem to be a little incremental though it achieves the best performance in most cases. It didn't deal with physical plausibility like 3D penetration. Its setting is also not unique as the main difference with previous method is changing from two-state to multi-state. The authors may elaborate on what's new that we are learning when building articulated objects.

Though I still have the mentioned concerns, I currently vote for borderline accept due to the extensive experiments and the SOTA performance.

**Questions:**

See above.

---

> ### Author Response · Authors · 2025-11-22
> **Response to Reviewer 5MT6**
>
> We appreciate the reviewer's positive assessment of our work, especially regarding our extensive experimental evaluations, the demonstrated state-of-the-art performance across diverse scenarios, and RS-Art dataset contribution.
>
> Here, we provide detailed responses to the reviewer's specific comments and suggestions:
>
> > **W1:** Complicated System.
>
> **A1:** We appreciate the reviewer's insightful comment. We think such modular components necessary and beneficial to effectively address this challenging task, without sacrificing extensibility or potential for further research.
> + Self-supervised modeling articulated objects with multiple movable parts is inherently ill-posed, characterized by significant structural and motion ambiguities. Even under simplified conditions where only a single part moves at a time (as shown in Fig. 3), purely end-to-end approaches still frequently fail, underscoring the benefit of introducing suitable modular priors.
> + Our core contribution is the construction of a unified Gaussian field representation for articulated objects. Importantly, the design space for obtaining this unified representation is highly flexible and open for extensive exploration, including potential variations, simplifications, or improvements to individual components or the overall pipeline.
>
> > **W2:** Details of RS-Art Dataset.
>
> **A2:** We thank the reviewer for the interest in the RS-Art dataset. We have provided detailed descriptions of the data collection, interaction states, modeling pipeline, and dataset structure in Appendix Sec. B. And Table 5 and Figure 13 explicitly show our evaluation results of our method on this dataset. We plan publicly release the dataset and provide more detailed descriptions of the dataset in the future, which we believe would further benefit this community.
>
> > **W3:** A little incremental method and what's new in learning articulated objects.
>
> **A3:** Our approach is neither simply an extension from two-state to multi-state inputs, nor merely a combination of existing components. Instead, we fundamentally reformulate articulated object modeling by introducing a unified, continuous, and part-aware Gaussian field. This novel representation directly addresses two critical limitations of prior methods: (1) existing geometry-first pipelines heavily rely on mesh quality, limiting their generalization; and (2) dynamic 3DGS approaches lack explicit part-level decoupling and controllable articulation. As illustrated in the provided [video](https://pd2gs-iclr.github.io/pd2gs-iclr/ ), our method achieves a tight unification of geometric deformation and realistic rendering.
>
> By leveraging this unified Gaussian field representation, our method learns several fundamentally new capabilities:
> + a continuous latent-conditioned deformation space enabling smooth interpolation and generation of novel intermediate states;
> + self-supervised discovery and decoupling of object parts, implicitly learning accurate kinematic structures;
> + a coherent spatial representation that integrates geometry and motion, avoiding representational fragmentation and enhancing interpretability and controllability.
>
> Collectively, these contributions provide substantial new insights and capabilities beyond existing methods for modeling articulated objects.
>
> > **W4:** How to deal with physical plausibility.
>
> **A3:** Our method avoids 3D penetration without relying on explicit geometric or collision constraints. Instead, the way we optimize the unified Gaussian field inherently encodes these properties:
>
> + Part-level decoupling: motion-based trajectory clustering naturally separates object parts and avoids inter-part collisions;
> + Latent-conditioned continuous deformation: the continuous latent deformation ensures structurally consistent and smooth transitions, preventing unnatural overlaps or penetrations;
> + SAM-guided boundary refinement: boundary-aware Gaussian splitting explicitly enforces clear, non-overlapping part boundaries.
>
> These mechanisms, inherently built into our unified Gaussian field optimization, naturally lead to physically plausible results. As demonstrated clearly in Figure 3 of the main paper and Figure 12 in the appendix, our approach consistently avoids penetration artifacts across various articulated types.

---

> > ### Comment · Reviewer_5MT6 · 2025-11-26
> >
> > Thanks for the reply. I still vote for accepting this paper.

---

### Official Review · Reviewer_VL5q · 2025-10-28

**Soundness:** 3
**Presentation:** 3
**Contribution:** 3
**Rating:** 6
**Confidence:** 4

**Summary:**

The paper proposes PD²GS, a self-supervised framework for articulated object modeling using 3D Gaussian Splatting. It learns a shared canonical Gaussian field and represents each interaction state as a continuous deformation via latent codes. A coarse-to-fine segmentation clusters Gaussian primitives by deformation trajectories and refines part boundaries using SAM-guided splitting, enabling part-level reconstruction and motion estimation. The authors also introduce RS-Art, a real-to-sim RGB-D dataset for evaluating generalization. Experiments show strong improvements over prior work on both synthetic and real objects.

**Strengths:**

- Technical contribution: the paper proposes a conceptually elegant unification of geometry and kinematics via continuous deformation of a canonical Gaussian field. Coarse-to-fine segmentation combining motion trajectories with SAM-driven boundary refinement is both novel and effective.

- RS-Art dataset is a meaningful contribution, bridging synthetic–real gaps with paired RGB-D data and 3D models.

- Comprehensive experiments on an expanded PartNet-Mobility split and the new dataset demonstrate strong performance and generalization.

**Weaknesses:**

- Pipeline is complex and involves many heuristic components, which limited the scalability of the method.
- The method proposed in the paper seems to require  multiple states, which puts forward more requirements for the data curation.  Furthermore, ensuring that the camera coordinate systems of all states are aligned is a challenge.  Outside the laboratory environment, such as in simple home scenarios, it is difficult for us to obtain states with multiple coordinate systems aligned, and the errors caused by coordinate misalignment are very likely to lead to failure.

**Questions:**

see weakness

---

> ### Author Response · Authors · 2025-11-22
> **Response to Reviewer VL5q 1/2**
>
> We thank the reviewer for their valuable feedback and for highlighting the strengths of our submission, particularly the conceptual elegance of unification of geometry and kinematics via our continuous deformation of a canonical Gaussian field, the effectiveness of coarse-to-fine segmentation strategy combined with SAM-driven refinement, and the meaningful contribution of the RS-Art dataset for evaluating generalization.
>
> Here, we provide detailed responses to the reviewer's specific comments and suggestions:
>
> > **W1:** Pipeline is complex and limited scalability.
>
> **A1:** We appreciate the reviewer's insightful comment. We consider the complexity of our pipeline essential and beneficial for this challenging task, without limiting scalability.
> + Self-supervised modeling of articulated multi-part objects is inherently an ill-posed problem, characterized by significant ambiguity in part composition and articulation patterns. Therefore, introducing effective priors (e.g., motion and semantic priors) is currently a highly practical strategy for addressing this challenge.
> + Our work evaluates a broader range of object categories and a greater number of object instances compared to previous approaches, including evaluations on real-world datasets. These comprehensive experiments demonstrate that our method consistently achieves superior performance, particularly for real-world articulated objects. Furthermore, while our pipeline includes multiple modules, it imposes no specific constraints that inherently limit scalability. To further confirm this, as shown in Tab. 1, we have provided additional results on objects with five articulated parts, clearly demonstrating that our method remains effective as complexity increases.
>
> *Table 1: Evaluation of our method on randomly selected object instances with five articulated parts. Each reported metric is averaged across individual parts per instance.*
> | **Object Instance** | **Method** | **Axis Ang ↓** | **Axis Pos ↓** | **Part Motion ↓** | **CD-s ↓** | **CD-m ↓** | **CD-w ↓** |
> |:---:|:---:|:---:|:---:|:---:|:---:|:---:|:---:|
> | Storage 45271 | PARIS | 61.36 | 0.41 | 48.79 | 5.86 | 380.66 | 4.96 |
> | | ArtGS | 1.60 | 0.01 | 1.94 | 2.45 | 4.27 | 1.77 |
> | | Ours | **0.02** | **0.00** | **0.02** | **1.39** | **0.71** | **1.25** |
> | Storage 45612 | PARIS | 61.37 | 0.14 | 46.16 | 5.89 | 577.17 | 5.07 |
> | | ArtGS | 0.11 | 0.08 | 1.82 | 2.53 | 4.09 | 1.83 |
> | | Ours | **0.04** | **0.03** | **0.01** | **1.40** | **0.53** | **1.32** |
> | Table 30666 | PARIS | 36.58 | - | 0.30 | 7.73 | 361.76 | 6.52 |
> | | ArtGS | 8.91 | - | 0.15 | 9.68 | 1159.34 | 5.70 |
> | | Ours | **0.15** | - | **0.05** | **4.05** | **1.47** | **3.65** |
> | Table 33810 | PARIS | 1.07 | 0.03 | **0.92** | **3.53** | 73.97 | 3.44 |
> | | ArtGS | 48.67 | 32.02 | 8.04 | 5.75 | 1346.58 | **3.18** |
> | | Ours | **0.44** | **0.02** | 0.93 | 4.85 | **0.68** | 3.91 |
>
> > **W2:** More requirements for data curation.
>
> **A2:** As shown in Appendix Table 4, our method achieves strong performance even with only two interaction states, the minimal common setup for this task. Moreover, our method uniquely supports inputs with an arbitrary number of interaction states, allowing users to flexibly adjust the number of states based on practical conditions and desired modeling quality.

---

> ### Author Response · Authors · 2025-11-22
> **Response to Reviewer VL5q 2/2**
>
> > **W3:** Camera pose aligned for multiple states.
>
> **A3:** In our RS-Art experiments (Appendix Table 5 and Figure 13), ground-truth camera poses were not provided. Given that articulated objects typically feature dominant static regions, camera poses estimated by COLMAP using its "image_registrator" achieved consistently accurate and stable alignment in our experiments, despite most real-world objects in our dataset being weakly textured and thus posing representative challenges for pose estimation. In the revised version, we will further include visualization results to clearly illustrate and validate the robustness of our approach.
>
> To evaluate robustness to camera pose inaccuracies, we conducted three experimental setups by adding Gaussian perturbations to the rotation and translation components of camera poses, as well as using pose estimations computed by COLMAP. As summarized in Table 2, although performance decreases slightly due to these perturbations, our method consistently maintains superior results compared to prior methods.
>
> *Table 2: Robustness evaluation under camera pose perturbations. Metrics are averaged across parts and objects, consistent with the setup in Table 1 of the main paper. In “(u, v)”, u denotes the standard deviation of rotation perturbation, and v denotes the standard deviation of translation perturbation.*
> | **Method** | **Axis Ang ↓** | **Axis Pos ↓** | **Part Motion ↓** | **CD-s ↓** | **CD-m ↓** | **CD-w ↓** |
> |:---:|:---:|:---:|:---:|:---:|:---:|:---:|
> | PARIS w/o Perturbation | 29.98 | 0.38 | 70.84 | 12.98 | 103.22 | 10.95 |
> | ArtGS w/o Perturbation | 3.53 |3.46|5.49| 4.29 | 82.65 | 6.72 |
> | Ours w/ (1.0°, 0.02 ) | 2.92  | 0.22 | 3.66 | 11.02 | 11.09 | 7.05 |
> | Ours w/ (3.0°, 0.05)  | 10.00 | 0.24 | 1.55 | 13.60 | 24.71 | 9.19 |
> | Ours w/ colmap | 1.93  | 0.11 | 3.85 | 7.39  | 6.82  | 7.63 |
> | Ours w/o Perturbation | **0.40**  | **0.06** | **0.43** | **2.05**  | **1.12**  | 3.02 |

---

> > ### Comment · Reviewer_VL5q · 2025-11-25
> >
> > Thank you for your reply; it has resolved my concerns, and I believe this work is acceptable for publication.

---

### Official Review · Reviewer_8QYT · 2025-10-31

**Soundness:** 2
**Presentation:** 2
**Contribution:** 2
**Rating:** 6
**Confidence:** 4

**Summary:**

This work presents PD$^2$GS, a framework for modeling articulated objects that overcomes the fragmentation and drift issues in existing self-supervised methods. It learns a shared canonical Gaussian field and represents arbitrary states as continuous deformations, jointly encoding geometry and kinematics. By associating each state with a latent code and using vision priors for part boundary refinement, PD$^2$GS enables accurate part-level decoupling while maintaining coherence. The method supports part-aware reconstruction, continuous control, and kinematic modeling without manual supervision.

**Strengths:**

1. The paper introduces a unified framework that models articulated objects through continuous deformations of a shared canonical Gaussian field, effectively addressing the fragmentation and drift issues inherent in previous discrete-state reconstruction methods.

2. The method achieves part-level decoupling without manual supervision by leveraging generic vision priors and latent code associations, enabling fine-grained continuous control over articulated configurations.

3. The paper contributes RS-Art, a valuable real-to-sim RGB-D dataset with reverse-engineered 3D models, facilitating rigorous evaluation on real-world data.

**Weaknesses:**

1. The reconstruction results exhibit excessive noise, particularly evident in the real-world examples shown in Figure 13, which raises concerns about the method's robustness in practical scenarios.

2. In Section 3.2 on deformable Gaussian splatting, the methodology bears strong similarity to existing 4DGS works such as [a], yet these related approaches are not cited or discussed.

3. The paper does not provide information about inference time per sample, which would be valuable for understanding the practical applicability of the method.

4. There are some related works that are missing in the paper: [b][c][d][e]

[a] 4D Gaussian Splatting for Real-Time Dynamic Scene Rendering;

[b] SINGAPO: Single Image Controlled Generation of Articulated Parts in Objects;

[c] Part2GS: Part-aware Modeling of Articulated Objects using 3D Gaussian Splatting;

[d] REACTO: Reconstructing Articulated Objects from a Single Video;

[e] NAP: Neural 3D Articulation Prior.

**Questions:**

Please see the weaknesses. I am hesitant about the rating primarily due to the reconstruction quality. Since this is fundamentally a reconstruction task, the results appear too coarse and do not meet the expected level of fidelity for such work.

---

> ### Author Response · Authors · 2025-11-22
> **Response to Reviewer 8QYT 1/2**
>
> We thank the reviewer for the valuable feedback and for recognizing several positive aspects of our submission, including the continuous deformations of a shared canonical Gaussian field, effective part-level decoupling without manual supervision, fine-grained continuous control, and the contribution of the real-to-sim RS-Art dataset for rigorous evaluation.
>
> Below, we provide detailed responses to the reviewer's primary comments and concerns:
>
> > **W1 & Q1:** Reconstruction quality in the real world.
>
> **A1:** Thanks for reviewer's insightful comment and fully agree that real-world articulated object reconstruction is a highly challenging problem.
> + We introduced the RS-Art dataset specifically to benchmark methods under highly challenging conditions (e.g., low-texture objects against controlled green-screen backgrounds). Under these conditions, all RGB-only baseline methods completely fail, except for ArtGS. In contrast, our method substantially improves performance across all evaluation metrics, demonstrating its robustness and effectiveness.
> + Benefiting from our unified Gaussian field representation, our approach, unlike previous methods, leverages inherent correlations encoded within the deformation field itself, enabling realistic rendering from arbitrary interaction states and viewpoints without relying on highly accurate geometric reconstructions (see Fig. 5 and Fig. 13 interpolation). This capability can be effectively utilized in AR/VR and simulation scenarios. As demonstrated in the [video](https://pd2gs-iclr.github.io/pd2gs-iclr/ ), our method simultaneously achieves realistic visual rendering during robot manipulation of articulated objects.
>
> >**W2:** Discussion for 4DGS works.
>
> **A2:** As noted by the reviewer, we have cited and discussed [a] along with other dynamic Gaussian splatting methods in our related work. Our approach significantly differs from these prior methods in both motivation and methodological design.
> + While methods such as [a] primarily target continuous-time dynamic scenes for real-time novel-view rendering, we specifically focus on articulated object modeling from discrete interaction states.
> + Methodologically, 4DGS works employs a deformation field parameterized by continuous time without explicitly modeling part-level structures or discrete articulation states. In contrast, our latent-code-driven deformation explicitly captures discrete interaction states and provides explicit part-level decoupling and fine-grained state control.
> + Our experiments demonstrate significantly improved performance in part segmentation accuracy, joint parameter estimation, and state interpolation, which are capabilities beyond the scope of these works.
>
> We will further clarify these distinctions explicitly in the revised introduction and related work sections.

---

> > ### Comment · Reviewer_8QYT · 2025-11-27
> >
> > Thank you for your response. My primary concern remains the reconstruction quality (**see Figure 13**) as mentioned in `Questions` section. While you argue that your experimental results demonstrate the method's `robustness and effectiveness`, I respectfully disagree. My concern about visual quality issues is not adequately resolved with your current response.

---

> > > ### Author Response · Authors · 2025-12-03
> > > **Further Clarification on Reconstruction Visual Quality**
> > >
> > > We appreciate the reviewer’s continued discussion and valuable feedback regarding the reconstruction visual quality. To further address this, we highlight two critical aspects:
> > >
> > > + Superior Robustness in Challenging Settings: Unsupervised 3D reconstruction solely from RGB inputs, without strong geometric priors, is inherently ill-posed. The RS-Art dataset presents significant challenges with limited texture details and unconstrained backgrounds. In these scenarios, most baseline methods fail to converge or yield valid outputs, and the few that do run suffer from catastrophic failures. In contrast, our method achieves substantial improvements and structural completeness, as demonstrated in Appendix D.3.
> > >
> > > + New Experiments Validating Real-World Practicality: To decisively demonstrate our method’s practical value and superiority, we conducted additional experiments on newly sampled real-scene data from RS-Art, as well as on instances provided by PARIS. As shown in Figure 15, even under complex lighting conditions in realistic environments, our method produces high-quality, practically usable reconstructions, maintaining clear advantages over previous state-of-the-art methods.

---

> ### Author Response · Authors · 2025-11-22
> **Response to Reviewer 8QYT 2/2**
>
> >**W3:** Inference time for per sample.
>
> **A3:** We appreciate the reviewer's insightful suggestion. As shown in Table 1 below, we provide additional experiments explicitly evaluating modeling and rendering efficiency. Specifically, we report the training time required to obtain the part-aware Gaussian field representation for objects with different numbers of movable parts (200 views per object), as well as inference time for rendering one view. The overall training time remains dominated by the core 3D Gaussian Splatting optimization process. Moreover, thanks to our unified Gaussian representation, inference speed after training remains consistent and efficient across all objects. We will include these details explicitly in the revised version.
>
> *Table 1: Training and inference time for objects with varying numbers of articulated parts.*
> | **Number of parts** | **Method** |  **2** | **3** | **5** |
> |:---:|:---:|:---:|:---:|:---:|
> | **Training (min)**  | PARIS | 31.38 | 45.99 | 75.15 |
> | | DTArt| 26.31 | 31.42 | 43.76 |
> | | ArtGS | 16.22| 15.08 | 17.86|
> | | Ours | 35.59 | 38.20 | 44.82 |
> | **Inference (ms)** | PARIS | 24773.60 | 37180.59 | 61924.30 |
> | | Ours | 37.93 | 40.86 | 37.22 |
>
>
> >**W4:** Some related works.
>
> **A4:** Most of the works mentioned address tasks substantially different from ours.
> + REACTO [d] focuses on reconstructing articulated objects from single videos without modeling explicit part-level geometry or providing controllable articulation states, thus differing significantly from our setting.
> + SINGAPO [b] and NAP [e] are conditional generative approaches, taking single images as input to synthesize plausible articulated shapes, rather than accurately modeling the geometry and motion of specific instances as in our task.
> + Part2GS [c] was unpublished and not publicly available when we conducted this work, and remains unavailable.
> + We have performed additional tests using the single-image input setting of methods [b] and [e] on the "Storage" category object. As shown in Table 2 below, these approaches show significant inaccuracies in analyzing articulated motion compared to our method.
>
> We will explicitly clarify these distinctions and further discuss these related works in our revision.
>
> *Table 2: Additional comparisons on the "Storage" objects.*
> | **Object** | **Method**  | **Axis Ang ↓** | **Axis Pos ↓** | **Part Motion ↓** | **CD-s ↓** | **CD-m ↓** | **CD-w ↓** |
> |:---:|:---:|:---:|:---:|:---:|:---:|:---:|:---:|
> | Storages | NAP [e] | 3.88 | 2.45 | 7.29 | 5.56 | 5.19 | 8.54 |
> | | SINGAPO [b] | 2.72 | 0.86 | 9.03 | – | – | – |
> | | Ours | **0.13** | **0.02** | **0.32** | **2.78** | **0.07** | **5.48** |

---

> > ### Comment · Reviewer_8QYT · 2025-11-27
> >
> > Thank you for your response. In Table 1, why do you have two `ours` for inference time?

---

> > > ### Author Response · Authors · 2025-11-27
> > >
> > > We apologize for the oversight. Specifically, in the inference row of Table 1, the first entry labeled as “Ours” should be “PARIS,” and only the second entry corresponds to our method. We have corrected this error in our revised response. Thank you for pointing out this issue.

---

> > > ### Comment · Reviewer_8QYT · 2025-11-27
> > >
> > > How about DTArt and ArtGS?

---

> > > > ### Author Response · Authors · 2025-12-03
> > > > **Regarding inference times for DTArt and ArtGS**
> > > >
> > > > As clarified in our paper, DTArt and ArtGS are designed for discrete states and cannot natively infer arbitrary continuous intermediate states, making a direct comparison infeasible. To provide these metrics, we explicitly deformed their reconstructed models using their inferred joint parameters. Notably, since ArtGS inherently lacks continuous deformation capabilities, we simulated intermediate states by rigidly transforming its reconstructed Gaussian field; therefore, its inference time is effectively equivalent to standard static 3DGS rendering.
> > > >
> > > > In the context of articulated object modeling, inference efficiency is not the primary limiting factor. Thus, this difference is secondary to the substantial gains in modeling capability.
> > > >
> > > > | **Number of parts** | **Method** |  **2** | **3** | **5** |
> > > > |:---:|:---:|:---:|:---:|:---:|
> > > > | **Inference (ms)** | PARIS | 24773.60 | 37180.59 | 61924.30 |
> > > > | | DTArt | 568.67 | 525.70 | 613.30 |
> > > > | | ArtGS | 3.42 | 3.71 | 4.16 |
> > > > | | Ours | 37.93 | 40.86 | 37.22 |

---

### Author Response · Authors · 2025-12-03
**General Response to Reviewers and AC 2/2**

Regarding the reviewers' remaining questions, we have provided detailed clarifications and additional experiments, receiving positive feedback summarized below:

1. **Reconstruction Quality in Challenging Scenes** (`Reviewer 8QYT`): While we regret that we were unable to continue the discussion with the reviewer regarding this issue, we have provided further clarifications and added experiments that clearly resolve this confusion. In our response, we clarified that the RS-Art dataset intentionally presents greater challenges compared to typical real-world scenes, especially regarding unsupervised 3D reconstruction without strong geometric priors. Despite these challenges, our method has consistently demonstrated substantial improvements over baseline methods. Moreover, we have included additional real-world experiments and visualizations in our revision, clearly showing that our method maintains structural integrity and produces practically usable reconstructions even where baselines suffer from catastrophic failures, as shown in the **Fig. 15** of the Appendix. Finally, we emphasized that even when geometric reconstructions face difficulties, our approach uniquely enables rendering of visually realistic images from arbitrary articulated states ([video](https://pd2gs-iclr.github.io/pd2gs-iclr/ )), a capability challenging for other methods to replicate.

2. **Pipeline Rationale and Scalability** (`Reviewers VL5q`, `5MT6`): We clarified that the introduction of certain modular constraints is essential for addressing the inherent ill-posedness and ambiguity of self-supervised articulated object modeling. Importantly, we emphasized that our unified Gaussian field formulation remains highly flexible and readily extensible for further research. Additionally, we provided extra experimental results on objects with a greater number of articulated parts to empirically demonstrate that these modular components do not negatively impact scalability, as shown in **Tab. 6** of the Appendix. Our clarifications and additional results received highly positive feedback from both reviewers.

3. **Core Capabilities and Distinctive Advantages** (`Reviewers 5MT6`, `HeoQ`): We clarified that our core innovation lies in fundamentally reformulating articulated object modeling via a unified, continuous, and part-aware Gaussian field. This representation explicitly addresses two major limitations of prior approaches: (1) existing geometry-first pipelines heavily rely on reconstruction quality, limiting their generalization; and (2) dynamic 3DGS approaches lack explicit part-level decoupling and controllable articulation. As illustrated in the [video](https://pd2gs-iclr.github.io/pd2gs-iclr/ ), our method achieves a tight unification of geometric deformation and realistic rendering. `Reviewer 5MT6` confirmed that our clarifications successfully resolved their concerns.

4. **Evaluation across Camera Pose Noise Scales** (`Reviewers VL5q`, `HeoQ`): We first clarified that our method achieved robust and stable performance on the RS-Art dataset, despite the absence of ground-truth camera poses. To further validate robustness, we conducted additional comprehensive experiments in the **Tab. 8**, including evaluations under synthetic camera pose perturbations and scenarios relying solely on COLMAP-estimated poses. `Reviewer VL5q` confirmed that these experiments sufficiently addressed their initial concerns.

5. **Comparison with Deformation-based and Generative Approaches** (`Reviewers 8QYT`, `HeoQ`): We have thoroughly analyzed the additional baselines suggested by `Reviewer 8QYT`, clearly clarifying the distinctions between these methods and ours, and explicitly explaining the rationale behind excluding some from comparisons. For methods that could be compared quantitatively, we carefully aligned experimental conditions and reported the results in **Tab. 9**, clearly demonstrating our method’s distinct advantages. To address `Reviewer HeoQ`’s specific concern regarding comparisons with dynamic NeRF and 3DGS methods, we added new experiments in the **Tab. 2** and **Fig. 6**, further demonstrating the clear superiority of our method.

Additionally, we have provided more detailed responses regarding technical concerns raised about inference time and occlusion in **Appendix D.6** and the **Limitations** section, respectively.

We sincerely thank all reviewers for their positive comments and constructive feedback, and have carefully integrated all suggestions into the **revised manuscript**, with changes clearly **highlighted in blue**.

---

### Author Response · Authors · 2025-12-03
**General Response to Reviewers and AC 1/2**

We sincerely thank all reviewers and the area chair for their insightful comments and valuable recommendations.

`Reviewer 8QYT` appreciated that our method effectively addresses the fragmentation and drift issues inherent in prior discrete-state reconstruction approaches.

`Reviewers VL5q` and `HeoQ` positively highlighted our approach as providing an elegant and conceptually clear representation for articulated object modeling, describing it as a major step toward high-fidelity digital twins.

`Reviewer 5MT6` recognized our comprehensive and extensive experiments as solid and convincing.

After discussion, Reviewers VL5q and 5MT6 reaffirmed their strong support and explicitly recommended the acceptance of our work. Reviewers 8QYT and HeoQ have not responded further as of the discussion deadline.

To support further assessment for AC, we have summarized all reviewers' primary feedback in the table below:

| **Reviewer Comments** | Reviewer 8QYT | Reviewer VL5q | Reviewer 5MT6 | Reviewer HeoQ |
|-----------------------|:-------------:|:-------------:|:-------------:|:-----------:|
| **Strengths** | | | | |
| Unified continuous Gaussian field representation | ✔ | ✔ |  | ✔ |
| Novel and conceptually elegant approach | ✔ | ✔ |  | ✔ |
| Effective Self-supervised part-level decomposition | ✔ | ✔ | ✔ | ✔ |
| Extensive evaluation (synthetic & real datasets) | ✔ | ✔ | ✔ |   |
| Superior Performance | ✔ | ✔ | ✔ | ✔ |
| RS-Art real-world dataset contribution | ✔ | ✔ | ✔ | ✔ |
| **Concerns & Clarifications** | | | | |
| Reconstruction quality concerns | ✘ |   |   |   |
| Numerous components of the pipeline |   | ✘ | ✘ |   |
| Further Clarification of Method’s Learned Capabilities and Innovations |   |   | ✘ | ✘ |
| Robustness to camera pose inaccuracies |   | ✘ |   | ✘ |
| Comparison with more baselines | ✘ |   |   | ✘ |
| **Initial Rating** | 6 | 6 | 6 | 6 |

We appreciate the reviewers' shared recognition of our core innovations, including the **unified Gaussian field representation** for modeling articulated objects, the effective **coarse-to-fine process**, and the **self-supervised part decomposition and motion analysis**. Reviewers also positively highlighted our **extensive experimental evaluation** demonstrating clear performance advantages, and acknowledged our newly proposed **RS-Art dataset** as a valuable resource for the community.

---

### Meta-Review · Area_Chair_LknB · 2026-01-03

**Summary:**

$PD^2GS$ presents a framework for modeling articulated objects with a unified, continuous, and part-aware Gaussian field. This method addresses fragmentation and drift issues in previous discrete-state approaches while unifying geometric deformation and realistic rendering.

The initial scores are all positive (6666). After reviewing the discussion, the AC believes most concerns are well addressed, but 8QYT notes the mediocre geometry quality in Figure 13, which aligns with the quantitative data in Table 5. However, this is not a decisive factor for rejecting the paper; it is not merely a mesh-based reconstruction work. Its main contributions include the VLM-guided fine-grained partwise decoupling procedure, joint recovery of shape, color, and kinematics in a Gaussian field, and the learned continuous state latent code for physically plausible state interpolation. Additionally, they have released a real captured dataset, RS-Art, which will greatly benefit the articulated modeling community. Therefore, the AC decides to accept this paper as a poster and encourages the authors to integrate all feedback into the final version.

Tips: Bold numbers generally refer to "best results," not "our results." These highlights should be removed from all tables or made consistent with red-colored cells.

**Reviewer Concerns:**

- Reviewer 8QYT: Most concerns are well addressed, but the reasons for the relatively poorer reconstruction quality (Figure 13, compared to DTArt) have not been directly answered. This explains why "Ours" does not outperform DTArt in several examples (Table 5, Storage 45194, Table 23372). The authors' response focuses more on the "superior performance on the real dataset RS-Art" instead.

- Reviewer VL5q: Concerns are well addressed.

- Reviewer 5MT6: Concerns are well addressed.

- Reviewer HeoQ: Concerns are well addressed.

**Reviewer Scores:**

- Reviewer 8QYT: As noted, "I am hesitant about the rating primarily due to the reconstruction quality." Since this concern is somewhat bypassed, this reviewer may lower their rating (4 → 6).

- Reviewer VL5q: Voted for acceptance on Nov 25 (6).

- Reviewer 5MT6: Voted for acceptance on Nov 26 (6).

- Reviewer HeoQ: May maintain the rating (6).

---

### Decision · Program_Chairs · 2026-01-26

Accept (Poster)